# Comparative efficacy and safety of traditional Chinese medicine injections in patients with transient ischemic attack: A systematic review and network meta-analysis

**Yunhao Yi[1], Guangheng Zhang[1], Shimeng Lv[1], Yuanhang Rong[1], Hui Liu[1], Ming Li[2]\***

**1** First Clinical Medical College, Shandong University of Traditional Chinese Medicine, Jinan, China, **2** Office of Academic Affairs, Shandong University of Traditional Chinese Medicine, Jinan, China

\* liming1207@126.com

## Abstract

### Objectives

Traditional Chinese medicine (TCM) injections are extensively utilized for the treatment of transient ischemic attack (TIA). However, it remains unclear which specific TCM injection exhibits superior efficacy. In this study, we conducted a network meta-analysis to compare the clinical efficacy and safety of various TCM injections in the treatment of TIA, with the aim of identifying the optimal treatment regimen.

### Design

We searched seven databases to collect information on nine TCM injections for the treatment of transient randomized controlled trials (RCTs) for the treatment of transient ischemic attacks were collected from the establishment to August 2023. The methodological quality and risk of bias were assessed using the RoB 2.0 evaluation tool, and reticulated Meta-analysis was performed using R software and Stata software.

### Results

We ultimately included 58 RCTs involving 5502 patients and comprising 9 TCM injections. In terms of improving the total effective rate, Shuxuetong injection (P-score = 0.69) was the most effective. In addition, Shuxuetong injection was most effective in lowering total cholesterol (P-score = 1.00) and triglyceride (P-score = 1.00) levels. Notably, Shuxuetong injection remained the most prominent in reducing fibrinogen (P-score = 0.91). However, among other blood hemorheology indicators, Dengzhanhuasu injection was the best regimen in reducing plasma viscosity (P-score = 1.00), whole blood viscosity (high shear rate) (P-score = 0.87), and whole blood viscosity (low shear rate) (P-score = 0.90). It was found that Yinxingyetiquwu injection (P-score = 0.72) was the most effective in reducing the incidence of cerebral infarction. In terms of safety, 22 studies reported adverse effects and descriptive analyses showed that the number of adverse effects of combination therapy was comparable to that of conventional therapy and that the safety profile was good.

**Data Availability Statement:** All relevant data are within the manuscript and its Supporting Information files.

**Funding:** This study was supported by a study on the Inheritance Studio of Traditional Chinese Medicine Master Zhang Canjia of State Administration of Traditional Chinese Medicine (no. [2010]59).

**Competing interests:** NO authors have competing interests

**Abbreviations:** VIP, China Science and Technology Journal Database; SinoMed, Chinese Biomedical Literature Database; CI, confidence interval; CINeMA, Confidence In Network Meta-Analysis; CNKI, China National Knowledge Infrastructure; CT, conventional treatment; MD, mean difference; MeSH, Medical Subject Headings; NMA, network meta-analysis; OR, odds ratio; PRISMA, Preferred Reporting Items for Systematic Reviews and Meta-analyses; RCTs, randomized controlled trials; ROB 2.0, Risk of Bias assessment tool 2.0; TC, total cholesterol; TG, triglyceride; TCM, traditional Chinese medicine; Wanfang, Wanfang Database.

## Conclusions

TCM injections in combination with CT may be a safe and effective intervention for patients with TIA, of which Shuxuetong injection, Dengzhanhuasu injection, and Yinxingyetiquwu injection may be more noteworthy. The quality of the literature included in the study was low, so further validation is needed with larger sample sizes, higher quality, and more rigorously designed RCTs.

## Systematic review registration

[PROSPERO], identifier [CRD42023443652].

## 1 Introduction

Transient ischemic attack (TIA) is a temporary and reversible neurological disorder that occurs due to insufficient blood flow to the brain, spinal cord, or retina [1]. It is characterized by symptoms that last only a few seconds or minutes but can last up to a maximum of 24 hours. Ancillary tests do not show any lesions responsible for the symptoms. TIA, as a high-risk warning sign of cerebral infarction, is a different stage from cerebral infarction, belonging to the dynamic process of ischemic brain injury. Study shows [2]. The incidence of cerebral infarction was 3.9% within 48 h of TIA onset, whereas the incidence of cerebral infarction increased to 14.6% within 3 months. Patients with TIA have a high recurrence rate of both short-term and long-term ischemic events [3] and even a mortality rate of up to 12% within 1 year, which severely affects the recovery of patients' neurological function and greatly reduces their quality of life [4]. The practical clinical management of TIA presents a crucial opportunity to prevent ischaemic stroke. Active treatment not only alleviates patients' symptoms but also prevents cerebral infarction, safeguarding brain function, improving quality of life, and reducing economic burden. Therefore, the treatment and prevention of TIA is getting more and more attention.

TIA and ischemic stroke share common risk factors such as smoking, hypertension, hyperlipidemia, and diabetes mellitus. The pathogenesis is not completely clear, mainly including the microembolism theory, hemodynamic alteration theory, etc. [5]. Microembolism is associated with the unstable dislodgement of atherosclerotic plaques or attached thrombi and arterial occlusion by cardiac thrombi. Hemodynamic changes are closely associated with severe stenosis of arteries in the vertebrobasilar system, mainly as a result of atherosclerosis. Currently, in response to its pathogenesis, the treatment of TIA on an international scale is mainly based on antiplatelet, anticoagulation and intravenous thrombolysis [6, 7]. Antiplatelet drugs are used for the treatment of noncardiogenic TIA, mainly aspirin and clopidogrel alone or in combination. However, long-term use can increase various bleeding risks, impairment of liver and kidney function, and gastrointestinal reactions [8] and may produce drug resistance [9]. Noncardiac TIA are predominantly anticoagulated and face the same risk of pathologic bleeding [10]. At the same time, patients are prone to relapse or even progress to stroke after stopping the drug [11]. The efficacy of intravenous thrombolysis and its evidence in patients with acute ischemic stroke in which TIA is the primary clinical manifestation is still insufficient [12]. It has been shown that intravenous thrombolysis within 24 h of a TIA episode leads to a higher propensity for symptomatic intracranial hemorrhage [13]. Therefore, we need more sophisticated treatment programs.

Chinese medicine possesses several advantages when it comes to treating diseases, such as its ability to target multiple factors, pathways, and links. As a Chinese medicine formulation, traditional Chinese medicine (TCM) injections primarily consist of Chinese medicine extracts. These injections have been widely used in the treatment of TIA in combination with conventional Western medicine [14], yielding remarkable clinical efficacy. One such example is the Yinxingyetiquwu injection, containing Ginkgo biloba extract as its primary component. Research has shown that Ginkgo biloba extract exhibits antioxidant properties, acts as a free radical scavenger, and functions as a platelet-activating factor inhibitor [15, 16]. Furthermore, TCM injections contain active ingredients that can effectively alleviate underlying risk factors like hypertension, diabetes, and hyperlipidemia [17]. However, there is a lack of network meta-analysis (NMA) of Chinese medicine injections for the treatment of TIA. The available NMAs are all for cerebral infarction, which has a different degree of pathologic damage than TIA and thus are not fully applicable [18]. Furthermore, there are limited studies directly comparing different TCM injections for TIA, making it difficult to make evidence-based decisions on their efficacy. NMA can address these limitations by combining direct and indirect evidence and comparing various interventions. It offers a comprehensive evaluation and ranking of interventions to identify their strengths and weaknesses. Therefore, this study employs reticulated meta-analysis to compare the efficacy and safety of different herbal injections for TIA treatment, aiming to provide evidence-based medical support for optimal drug selection in clinical practice.

## 2 Methods

We performed a systematic review and NMA according to the Preferred Reporting Items for Systematic Reviews and Meta-analyses (PRISMA) statement [19]. In addition, this study has been registered with PROSPERO under the number CRD42023443652. Details of the PRISMA Checklist are provided in S1 File.

### 2.1 Search strategies

We searched the data in PubMed, Embase, Cochrane Library, China National Knowledge Infrastructure (CNKI), Wanfang Database (Wanfang), China Science and Technology Journal Database (VIP), and Chinese Biomedical Literature Database (SinoMed) from the database's inception through August 2023 using Medical Subject Headings (MeSH) for "transient ischemic attack" and "injections" search terms in S2 File. The following terms were primarily used in the search process: (transient ischemic attack, transient cerebral ischemia, TIA) and (injection, injectable). In order to ensure the comprehensiveness of the study, we conducted additional searches by reviewing the reference lists of previously published systematic reviews that were identified through the Cochrane Database of Systematic Reviews (search terms: transient ischemic attack, injections; limits: none) and PubMed (search terms: transient ischemic attack, injections; limits: systematic reviews or meta-analysis). We also searched the Chinese Clinical Trial Registry and Clinicaltrials.gov for some unpublished clinical trials.

### 2.2 Inclusion standards

The inclusion criteria were based on the PICOS (participants, interventions, comparators, outcomes, and study design) approach [19]. Studies included in this meta-analysis must meet the following criteria and report specific experimental characteristics: (a) Participants must meet diagnostic criteria for TIA [20, 21] and have no responsible lesions on imaging. There were no restrictions on the age, gender, race, geographic region, ethnicity, or disease duration of the participants. (b) The intervention must involve the use of one of the nine TCM injections in

combination with conventional treatment (CT) (S3 File). The dosage and duration of the intervention were not limited, but they had to comply with the specified guidelines for their use. (c) The CT group must receive CTs from Western medicine, including antiplatelet, anticoagulation, and other CTs. They should not receive TCM injections, proprietary Chinese medicines, or soups. Symptomatic treatments such as antihypertensive, lipid-lowering, glucose-lowering, maintaining electrolyte balance, and nutritive neurological treatment can be taken appropriately for those with underlying diseases. (d) The study must include at least one outcome indicator. (e) The study design of the included articles must be a randomized controlled trial, and there were no language restrictions.

## 2.3 Exclusion standards

(a) studies with duplicate publications or duplicate data; (b) non-RCT studies, such as meta-analyses, reviews, theoretical discussions, clinical experiences, animal experiments, etc.; (c) the literature could not be accessed in full text and the data could not be extracted; (d) studies that did not have one primary endpoint or secondary endpoint indicator as a primary endpoint indicator; (e) studies in which the interventions included non-conventional treatments, such as other proprietary Chinese medicines for the treatment of transient ischemic attacks, herbal broths, acupuncture, gua sha; and (f) the relevant literature on injectable fluids was less than 3 articles.

## 2.4 Types of outcome measures

The study refers to domestic and international clinical guidelines and expert consensus for the treatment of TIA [6, 7, 22]. The total effective rate of this study was determined according to the criteria established by the Chinese Medical Association, [22] significantly effective: within 2 weeks of treatment, ischemic attacks were utterly controlled, and there was no recurrence within 1 month of follow-up; effective: within 2 weeks of treatment, the condition was under better control, and the number and duration of ischemic attacks were reduced by more than 50% compared with the previous one; ineffective: within 2 weeks of treatment, the number of ischemic attacks was unchanged or increased, and the duration of the attacks was prolonged or there were cerebral infarcts. Slow blood flow due to stenosis or microthrombosis further alters blood rheology, resulting in elevated whole blood viscosity, plasma viscosity, and fibrinogen. Therefore, some of the blood rheology indexes were selected as the outcome indexes of this study. Elevated lipids are an essential risk factor for ischemic stroke and were therefore included in the study.

Leading outcome indicators: (a) total effective rate (significantly effective + effective); (b) plasma viscosity; (c) fibrinogen: (d) whole blood reduced viscosity (high shear rate); (e)whole blood reduced viscosity (low shear rate). Secondary ending indicator: (a) total cholesterol (TC); (b) triglyceride (TG); (c) incidence of cerebral infarction; (d) Adverse events include rash, petechiae, dizziness, nausea, etc.

## 2.5 Literature screening and data extraction

Two researchers (YYH and ZGH) conducted a literature search and retrieved relevant articles based on the predetermined search criteria. They utilized Endnote X9 to screen and remove duplicate publications. The researchers then applied specific inclusion criteria to assess the literature independently. Firstly, they excluded studies that did not meet the requirements based on the title and abstract. Subsequently, they thoroughly reviewed the full texts to identify randomized controlled trials (RCTs) suitable for quantitative analysis. Relevant information from

the included studies, such as study details, characteristics, and outcome indicators, was extracted using Excel. In cases of disagreement, a third researcher (LH) was consulted.

## 2.6 Risk of bias assessment and CINeMA

Two investigators (YYH and ZGH) referred to the Cochrane Collaboration's recommendation of the latest Risk of Bias assessment tool 2.0 (ROB 2.0) for risk of bias assessment [23]. ROB 2.0 comprises five modules: randomization process, deviations from intended interventions, missing outcome data, measurement of the outcome, and selection of the reported result. The results of each module were assessed using the modular decision pathway diagrams. Ultimately, these results were summarized to determine the overall assessment of bias, which was categorized as "Low risk," "Some concerns," or "High risk" based on the contents of the literature. CINeMA (Confidence In Network Meta-Analysis: https://cinema.ispm.unibe.ch/) is a web tool that expands the capabilities of GRADE (Grading of Recommendations Assessment, Development, and Evaluation). We used the CINeMA to assess the certainty of evidence for each outcome, categorizing the evidence into four levels: high, moderate, low, and very low.

## 2.7 Statistical investigation

We used R software (version 4.3.2) and Stata software (version 14.0) for statistical analysis. Within a frequentist framework, we used the "netmeta" package in R for NMA analysis. We visualized the evidence network, where the size of the points represented the sample size of the intervention, and the thickness of the connecting lines indicated the amount of evidence for direct comparisons between the two interventions. We made pooled estimates tables of the results of the network meta-analyses to determine whether the two-by-two comparisons were statistically significant or not based on effect values and confidence intervals. Dichotomous variables were presented as odds ratio (OR), and continuous variables were presented as mean difference (MD), along with their respective 95% confidence interval (CI). If the 95% CI of the OR contained 1 or the 95% CI of the MD contained 0, the difference between the two-by-two comparisons of the interventions was not statistically significant. We assessed overall heterogeneity using the "decomp.design" function in the R software, and heterogeneity was expressed as $I^2$. If the $I^2$ value exceeded 50%, significant heterogeneity was indicated, necessitating the application of a random effects model. Global inconsistency analyses and node split analyses were conducted to assess the discordance between direct and indirect evidence estimates for each intervention comparison, typically indicated by *p*. The forest plots depict the summary values of MD or OR for all comparisons along with their 95% CI, and the P-scores assess the efficacy of various TCM injections, with higher scores indicating superior efficacy. Additionally, a cluster analysis of the two distinct outcome metrics was conducted to identify interventions with improved combined efficacy. To identify sources of heterogeneity, meta-regression analysis was conducted. We conducted sensitivity analyses limited by the time or number of people in the intervention to test the stability of the results. We plotted comparison-adjusted funnel plots to see if studies were evenly distributed on both sides of the center line to identify small-sample effects.

## 3 Results

### 3.1 Literature screening process and essential characteristics

A total of 1119 documents were obtained by searching relevant databases, and the titles and abstracts of 853 documents were obtained by excluding 266 duplicates through Endnote X9 software. After reading the titles and abstracts of the literature, 755 apparently non-compliant

literature, such as studies of non-RCTs, theoretical explorations, and interventions that did not meet the requirements, were excluded; 98 selected literature were read in full for further assessment. In accordance with the inclusion and exclusion criteria, 40 pieces of literature were excluded, including 8 studies that did not contain the included outcome indicators, 17 studies with non-compliant interventions, 12 studies with less than 3 studies of the same herbal injection included in the study, and 3 studies with incomplete data. Fifty-eight studies of RCTs were finally included, all of which were in the Chinese literature. The literature screening process is shown in Fig 1.

A total of 58 studies [24–81] of RCTs were included in this study, which was published between 2005 and 2023, involving 5502 patients, 2761 receiving herbal injections in combination with CT, and 2741 receiving CT, involving nine types of herbal injections, including Danhong injection 10 items [24–33], Xuesaitong injection 3 items [34–36], Xueshuantong injection 7 items [37–43], Dengzhanhuasu injection 6 items [44–49], Shuxuening injection 5 items [50–54], Guhong injection 3 items [55–57], Shuxuetong injection 13 items [58–70], Shenxiongputao injection 4 items [71–74], Yinxingyetiquwu injection 7 items [74–81]. The authors of the included literature, time of publication, subject information (mean age, gender), intervention, course of treatment, treatment characteristics, and essential characteristics of outcome indicators are shown in Table 1.

### 3.2 Bias risk assessment of involved literature

The risk of bias in 58 studies [24–81] was evaluated using ROB 2.0. Two studies [40, 43] were grouped in the order of visit, 14 studies [24, 25, 27, 39–42, 46, 55, 60, 61, 67, 74, 77] employed the random number table method, 2 studies [28, 30] employed lottery randomization, 2 studies [34, 35] utilized randomized numbering, 38 studies [26, 29, 31–33, 36–38, 43–45, 47, 48, 50, 51, 53, 54, 56–59, 62–66, 68–73, 75, 76, 78–81] mentioned "randomization" without explicitly describing the specific method employed. None of the 58 studies mentioned allocation concealment. Fifty-eight studies reported comparable baseline information between the groups. None of the 58 studies mentioned double-blind. Regarding the results section, 49 studies [24–30, 32–36, 38–43, 46–51, 53, 54, 56, 59–65, 67–81] used a table combined with text to record the study participants' outcome indicators, and 9 studies [31, 37, 44, 45, 52, 55, 57, 58, 66] used textual descriptions; none had missing outcome data. 58 studies [24–81] described in detail the methods used to measure outcome indicators, and there were no multiple methods or multiple ways of analyzing outcome measures. The risk of bias assessment is detailed in Fig 2 and S4 File.

### 3.3 Outcome indicators

**3.3.1 Total effectiveness rate.** Forty-eight studies [24, 25, 27, 30–37, 43–69, 71–74, 76–81] (82.76%) and 4,476 participants (81.35%) assessed the total effective rate (Fig 3A). All 9 TCM injections combined with CT significantly improved the total effective rate compared to CT (Fig 4A). Ranked by the degree of change in total effectiveness rate, Shuxuetong injection + CT (P-score = 0.69) was the best and CT (P-score = 0) was the worst (Fig 4A). NMA results showed that there was no significant difference in any of the comparisons between different TCM injections combined with CT (S5 File).

**3.3.2 Plasma viscosity.** Twenty-four studies [29, 30, 34, 46, 49, 53–56, 60, 64, 68, 70–81] (41.38%) and 2258 participants (41.04%) assessed the plasma viscosity (Fig 3B). 4 TCM injections combined with CT significantly improved the plasma viscosity compared to CT (Fig 4B). Dengzhanhuasu injection (MD = -1.09, 95% CI [-1.43; -0.76]), Yinxingyetiquwu injection (MD = -0.37, 95% CI [-0.53; -0.20]), Shenxiongputao injection (MD = -0.36, 95% CI [-0.58;

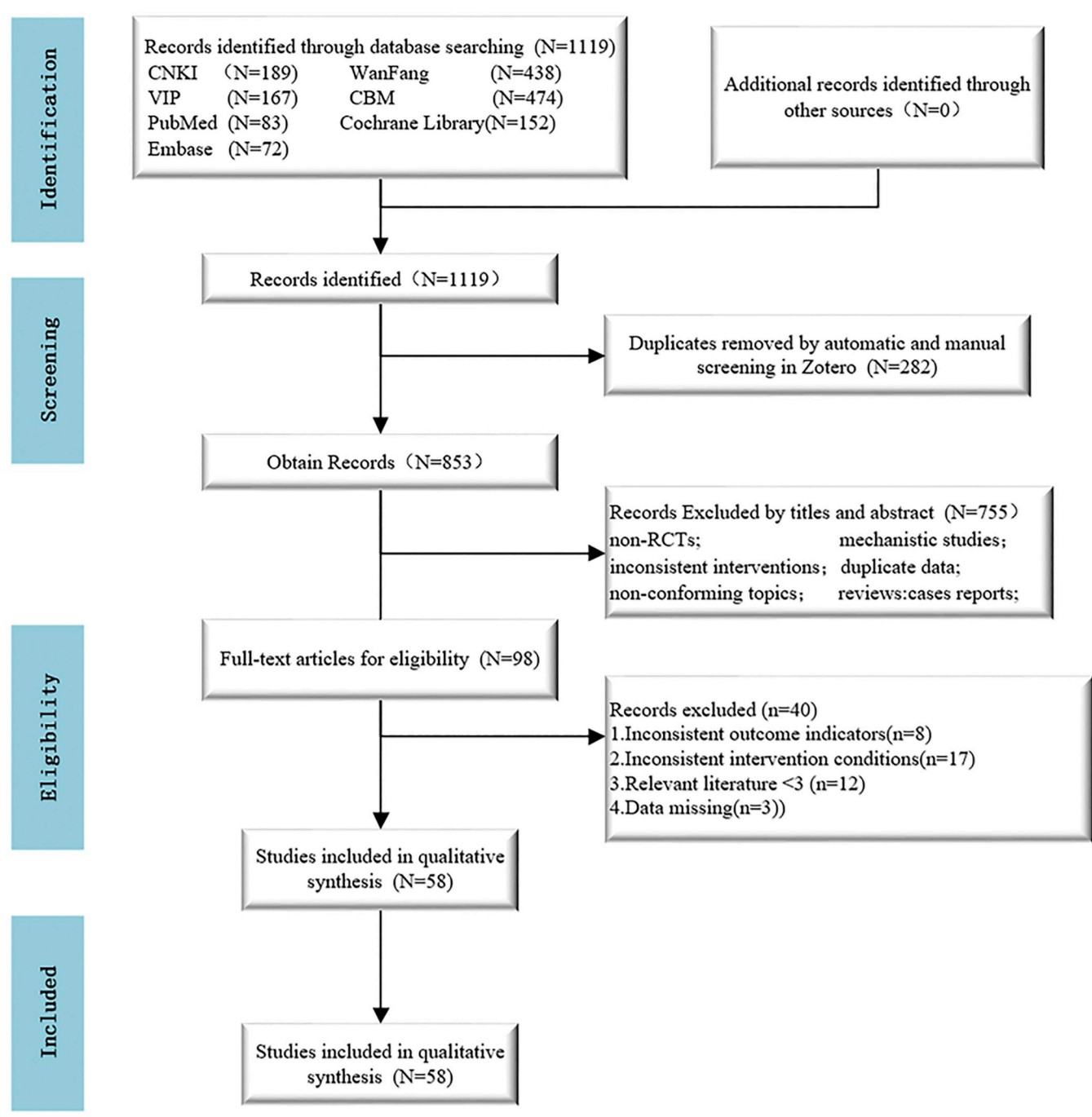

**Fig 1. Study selection process.**

-0.14]) and Guhong injection (MD = -0.36, 95% CI [-0.68; -0.04]) respectively in combination with CT were superior to CT. Ranked by the degree of change in plasma viscosity, Dengzhanhuasu injection + CT (P-score = 1.00) was the best and CT (P-score = 0.07) was the worst (Fig 4B). NMA results showed that Dengzhanhuasu injection + CT was superior to Yinxingyetiquwu injection + CT (MD = -0.73, 95% CI [-1.10; -0.35]), Shenxiongputao injection + CT (MD = -0.74, 95% CI [-1.14; -0.33]), Guhong injection + CT (MD = -0.73, 95% CI [-1.19;

**Table 1. Characteristics of included studies.**

| Included studies | Sample (E/C) | Sex (M/F) E | Sex (M/F) C | Age E | Age C | Intervention E | Intervention C | Disease duration | Outcomes |
|---|---|---|---|---|---|---|---|---|---|
| Zhang Youmei 2022 [24] | 30/30 | 16/14 | 17/13 | 58.41 ±11.23 | 58.36 ±11.33 | Conventional treatment, Danhong injection 10ml + 5% Glucose Injection 500ml, ivdrip, qd. | Conventional treatment | 1W | Total effective rate |
| Ruan Chunyun 2019 [25] | 72/72 | 44/28 | 40/32 | 69.07 ±3.39 | 68.93 ±3.24 | Conventional treatment, Danhong injection 40ml + 5% Glucose Injection 250ml, ivdrip, qd. | Conventional treatment | 2W | Total effective rate Adverse events |
| Cao Yinghua 2017 [26] | 43/43 | 22/21 | 23/20 | 59.16 ±4.38 | 60.02 ±4.47 | Conventional treatment, Danhong injection 40ml + 0.9% Sodium Chloride Injection 250ml, ivdrip, qd. | Conventional treatment | 2W | Incidence of cerebral infarction |
| Wang Lei 2016 [27] | 40/40 | 24/16 | 23/17 | 64.81 ±8.92 | 65.03 ±8.74 | Conventional treatment, Danhong injection 20ml + 0.9% Sodium Chloride Injection 250ml, ivdrip, qd. | Conventional treatment | 15D | Total effective rate |
| Deng Yuwen 2016 [28] | 43/43 | 51/35 | | 63.03±5.47 | | Conventional treatment, Danhong injection 40ml + 5% Glucose Injection 250ml, ivdrip, qd. | Conventional treatment | 2W | Incidence of cerebral infarction |
| Zhang Shujing 2015 [29] | 45/45 | 25/20 | 24/21 | 62±11.2 | 61.2 ±11.4 | Conventional treatment, Danhong injection 40ml + 0.9% Sodium Chloride Injection 250ml, ivdrip, qd. | Conventional treatment | 2W | Plasma viscosity Whole blood reduced viscosity (high shear rate) Whole blood reduced viscosity (low shear rate) |
| Kang Yonggang2015 [30] | 40/40 | 23/17 | 21/19 | 56.87 +4.11 | 56.98 +3.89 | Conventional treatment, Danhong injection + 5% Glucose Injection 250ml, ivdrip, qd. | Conventional treatment | 2W | Total effective rate Plasma viscosity Whole blood reduced viscosity (high shear rate) Whole blood reduced viscosity (low shear rate) TC, TG |
| Wang Chengzhang 2012 [31] | 30/30 | 29/31 | | NA | NA | Conventional treatment, Danhong injection 20ml + 0.9% Sodium Chloride Injection 100ml, ivdrip, bid. | Conventional treatment | 2W | Total effective rate |
| Chen Pengqi 2015 [32] | 31/27 | 17/14 | 15/12 | 62 | 63 | Conventional treatment, Danhong injection 40ml + 0.9% Sodium Chloride Injection, ivdrip, qd. | Conventional treatment | 21D | Total effective rate Adverse events |
| Zhao Li'an 2013 [33] | 82/82 | 51/31 | 48/34 | 56±2 | 55±2 | Conventional treatment, Danhong injection 30ml + 0.9% Sodium Chloride Injection250ml, ivdrip, qd. | Conventional treatment | 2W | Total effective rate Adverse events |
| Zhang Pengpeng 2020 [34] | 49/49 | 28/21 | 26/23 | 56.86 ±4.59 | 56.31 ±4.62 | Conventional treatment, Xuesaitong injection200-400ml + 5% Glucose Injection 250-500ml, ivdrip, qd. | Conventional treatment | 2W | Total effective rate Plasma viscosity Fibrinogen Whole blood reduced viscosity (high shear rate) Whole blood reduced viscosity (low shear rate) |
| Shi Min 2017 [35] | 49/49 | 63/35 | | 63.5 | | Conventional treatment, Xuesaitong injection 400ml + 5% Glucose Injection 250ml, ivdrip, qd. | Conventional treatment | 15D | Total effective rate Incidence of cerebral infarction |
| Chen Xinguang 2014 [36] | 34/34 | 20/14 | 21/13 | 45~75 | 48~75 | Conventional treatment, Xuesaitong injection 400ml + 0.9% Sodium Chloride Injection 250ml, ivdrip, qd. | Conventional treatment | 10D | Total effective rate Incidence of cerebral infarction |
| Chen Wanjiao 2011 [37] | 35/35 | 22/13 | 24/11 | 52.85 | 53.65 | Conventional treatment, Xueshuantong injection 400ml, ivdrip, qd. | Conventional treatment | 10D | Total effective rate |
| Xue Qinghua 2018 [38] | 34/34 | 18/16 | 21/13 | 65.9 ±6.2 | 64.3 ±6.1 | Conventional treatment, Xueshuantong injection 450ml, ivdrip, qd. | Conventional treatment | (5~6)*10D | Plasma viscosity TC, TG |
| Jing Yaping 2020 [39] | 40/40 | 25/15 | 26/14 | 64.35 ±5.11 | 63.94 ±5.39 | Conventional treatment, Xueshuantong injection 500ml, ivdrip, qd. | Conventional treatment | 6*10D | Fibrinogen TC, TG |

(*Continued*)

**Table 1.** (Continued)

| Included studies | Sample (E/C) | Sex (M/F) E | Sex (M/F) C | Age E | Age C | Intervention E | Intervention C | Disease duration | Outcomes |
|---|---|---|---|---|---|---|---|---|---|
| Wang Zhijun 2022 [40] | 50/50 | 27/23 | 29/21 | 68.46 ±3.46 | 68.59 ±3.52 | Conventional treatment, Xueshuantong injection 450ml, ivdrip, qd. | Conventional treatment | (5~6)*10D | Whole blood reduced viscosity (high shear rate) TC, TG Adverse events |
| Chen Dongyi 2021 [41] | 140/140 | 79/61 | 88/52 | 64.35 ±7.60 | 63.50 ±8.06 | Conventional treatment, Xueshuantong injection 450ml, ivdrip, qd. | Conventional treatment | (5~6)*10D | Fibrinogen TC, TG |
| Wu Yong 2020 [42] | 36/36 | 18/18 | 20/16 | 57.1 ±8.7 | 56.2 ±8.1 | Conventional treatment, Xueshuantong injection 250ml, ivdrip, qd. | Conventional treatment | 6*10D | Whole blood reduced viscosity (low shear rate) TG |
| Ji Qun 2011 [43] | 58/58 | 43/15 | 38/20 | 67.5 ±10.2 | 63.6 ±8.8 | Conventional treatment, Xueshuantong injection 250ml, ivdrip, qd. | Conventional treatment | 2W | Total effective rate |
| Qi Ji 2017 [44] | 47/47 | 26/21 | 27/20 | 57.8 ±3.1 | 57.5 ±3.2 | Conventional treatment, Dengzhanhuasu injection 40mg + 5% Glucose Injection 250ml, ivdrip, qd. | Conventional treatment | 2W | Total effective rate |
| An Meihua 2017 [45] | 54/54 | 30/24 | 31/23 | 54.8 ±3.3 | 54.5 ±3.3 | Conventional treatment, Dengzhanhuasu injection40mg + 5% Glucose Injection 250ml, ivdrip, qd. | Conventional treatment | 2W | Total effective rate |
| Gao Yuhong 2016 [46] | 62/62 | 41/21 | 38/24 | 56.3 ±10.3 | 57.6 ±10.7 | Conventional treatment, Dengzhanhuasu injection40mg + 5% Glucose Injection 250ml, ivdrip, qd. | Conventional treatment | 2W | Total effective rate Plasma viscosity Fibrinogen Whole blood reduced viscosity (high shear rate) Whole blood reduced viscosity (low shear rate) Incidence of cerebral infarction |
| Wang Qin 2017 [47] | 50/50 | 30/20 | 28/22 | 60.1 ±3.9 | 59.8 ±3.7 | Conventional treatment, Dengzhanhuasu injection40mg + 5% Glucose Injection 250ml, ivdrip, qd. | Conventional treatment | 2W | Total effective rate |
| Jiang Deping 2017 [48] | 53/53 | 32/21 | 30/23 | 57.48 ±4.16 | 56.89 ±4.02 | Conventional treatment, Dengzhanhuasu injection 40mg + 5% Glucose Injection 250ml, ivdrip, qd. | Conventional treatment | 2W | Total effective rate Incidence of cerebral infarction |
| Zhu Xuefei 2017 [49] | 25/25 | 31/19 | | 73.28±5.58 | | Conventional treatment, Dengzhanhuasu injection 40mg + 5% Glucose Injection 250ml, ivdrip, qd. | Conventional treatment | 2W | Total effective rate Plasma viscosity Fibrinogen Whole blood reduced viscosity (high shear rate) Whole blood reduced viscosity (low shear rate) |
| Zhang Min 2020 [50] | 25/25 | 16/09 | 15/10 | 51±2.0 | 50±2.3 | Conventional treatment, Shuxuening injection 20ml + 5% Glucose Injection 250ml, ivdrip, qd. | Conventional treatment | 2W | Total effective rate |
| Zhang Shimin 2015 [51] | 32/30 | 20/12 | 19/11 | 61.3 ±5.7 | 60.6 ±6.4 | Conventional treatment, Shuxuening injection 20ml + 5% Glucose Injection 250ml, ivdrip, qd. | Conventional treatment | 7D | Total effective rate Adverse events |
| Yu Yunqi 2005 [52] | 32/32 | 20/12 | 18/14 | 52±8.3 | 56.9 ±10.4 | Conventional treatment, Shuxuening injection 20ml + 5% Glucose Injection 250ml, ivdrip, qd. | Conventional treatment | 10~15D | Total effective rate |
| Chai Xuesen 2016 [53] | 46/46 | 27/19 | 26/20 | 59.1 ±11.2 | 48.5 ±12.7 | Conventional treatment, Shuxuening injection 20ml + 5% Glucose Injection 250ml, ivdrip, qd. | Conventional treatment | 1W | Total effective rate. Plasma viscosity Fibrinogen Whole blood reduced viscosity (high shear rate) Whole blood reduced viscosity (low shear rate). Incidence of cerebral infarction |

(*Continued*)

**Table 1.** (Continued)

| Included studies | Sample (E/C) | Sex (M/F) | | Age | | Intervention | | Disease duration | Outcomes |
|---|---|---|---|---|---|---|---|---|---|
| | | E | C | E | C | E | C | | |
| Le Mingjun 2014 [54] | 31/31 | 19/12 | 18/13 | 62.3 | 63.2 | Conventional treatment, Shuxuening injection 20ml + 5% Glucose Injection 250ml, ivdrip, qd. | Conventional treatment | 2W | Total effective rate Plasma viscosity Whole blood reduced viscosity (high shear rate) Whole blood reduced viscosity (low shear rate) TC Incidence of cerebral infarction |
| Zhang Kun 2017 [55] | 68/68 | 40/28 | 38/30 | 54.3 ±5.6 | 54.5 ±6.4 | Conventional treatment, Guhong injection 20ml + 0.9% Sodium Chloride Injection 250ml, ivdrip, qd. | Conventional treatment | 15D | Total effective rate Plasma viscosity Fibrinogen Whole blood reduced viscosity (high shear rate) Whole blood reduced viscosity (low shear rate) |
| Xue Xiaoxian 2018 [56] | 46/46 | 27/19 | 28/18 | 62.03 ±5.37 | 61.89 ±5.45 | Conventional treatment, Guhong injection 20ml + 0.9% Sodium Chloride Injection 250ml, ivdrip, qd. | Conventional treatment | 2W | Total effective rate Plasma viscosity Fibrinogen Whole blood reduced viscosity (high shear rate) Whole blood reduced viscosity (low shear rate) |
| Long Runbo 2019 [57] | 46/46 | 26/20 | 25/21 | 46.45 ±3.28 | 46.88 ±3.09 | Conventional treatment, Guhong injection 15ml + 0.9% Sodium Chloride Injection 250ml, ivdrip, qd. | Conventional treatment | 2W | Total effective rate |
| Zhang Meijing 2017 [58] | 42/42 | 22/20 | 23/19 | 56.3 ±2.7 | 56.1 ±3.0 | Conventional treatment, Shuxuetong injection 4ml + 0.9% Sodium Chloride Injection 250ml, ivdrip, qd. | Conventional treatment | 2W | Total effective rate |
| Yao Bo 2016 [59] | 66/66 | 34/32 | 35/31 | 56.68 ±8.59 | 56.73 ±8.62 | Conventional treatment, Shuxuetong injection 6ml + 0.9% Sodium Chloride Injection 250ml, ivdrip, qd. | Conventional treatment | 2W | Total effective rate Adverse events |
| Yang Huan 2016 [60] | 47/47 | 30/17 | 31/16 | 54 ± 5 | 53 ± 6 | Conventional treatment, Shuxuetong injection 6ml + 0.9% Sodium Chloride Injection 250ml, ivdrip, qd. | Conventional treatment | 2W | Total effective rate Plasma viscosity Fibrinogen Whole blood reduced viscosity (high shear rate) Whole blood reduced viscosity (low shear rate) |
| Sun Jiming 2013 [61] | 35/35 | NA | NA | 55.30 ±3.70 | 53.7 ±4.2 | Conventional treatment, Shuxuetong injection 6ml + 0.9% Sodium Chloride Injection 250ml, ivdrip, qd. | Conventional treatment | 2W | Total effective rate Whole blood reduced viscosity (high shear rate) Whole blood reduced viscosity (low shear rate) Incidence of cerebral infarction |
| Fan Jian 2012 [62] | 55/55 | 62/48 | | 56.9 ±7.3 | 57.2 ±5.7 | Conventional treatment, Shuxuetong injection 10ml + 0.9% Sodium Chloride Injection 250ml, ivdrip, qd. | Conventional treatment | 2W | Total effective rate Incidence of cerebral infarction |
| Luo Hongyong 2010 [63] | 44/42 | 24/20 | 22/20 | 43~72 | 41~73 | Conventional treatment, Shuxuetong injection 6ml + 0.9% Sodium Chloride Injection 250ml, ivdrip, qd. | Conventional treatment | 6*15D | Total effective rate Incidence of cerebral infarction |
| Chen Chibang 2009 [64] | 35/35 | 19/16 | 18/17 | 59.6 | 59.5 | Conventional treatment, Shuxuetong injection 10ml + 0.9% Sodium Chloride Injection 250ml, ivdrip, qd. | Conventional treatment | 2W | Total effective rate Plasma viscosity Whole blood reduced viscosity (high shear rate) |

(*Continued*)

**Table 1.** (Continued)

| Included studies | Sample | Sex (M/F) | | Age | | Intervention | | Disease duration | Outcomes |
|---|---|---|---|---|---|---|---|---|---|
| | (E/C) | E | C | E | C | E | C | | |
| Zhang Hua 2008 [65] | 72/64 | 39/33 | 36/28 | 56 | 54 | Conventional treatment, Shuxuetong injection 6ml + 0.9% Sodium Chloride Injection 250ml, ivdrip, qd. | Conventional treatment | 2W | Total effective rate Incidence of cerebral infarction |
| Che Dan 2018 [66] | 47/47 | 26/21 | 25/22 | 60.3 ±4.7 | 59.9 ±4.5 | Conventional treatment, Shuxuetong injection 4ml + 0.9% Sodium Chloride Injection 250ml, ivdrip, qd. | Conventional treatment | 2W | Total effective rate Incidence of cerebral infarction |
| Liang Zhongwei 2012 [67] | 50/50 | 30/20 | 35/15 | 57.3± 2.3 | 58.0 ±1.7 | Conventional treatment, Shuxuetong injection 6ml + 0.9% Sodium Chloride Injection 250ml, ivdrip, qd. | Conventional treatment | 2W | Total effective rate |
| Lin Feihong 2014 [68] | 51/51 | 28/23 | 27/24 | 65 | 66 | Conventional treatment, Shuxuetong injection 6ml + 0.9% Sodium Chloride Injection 250ml, ivdrip, qd. | Conventional treatment | 7D | Total effective rate Plasma viscosity Whole blood reduced viscosity (high shear rate) Whole blood reduced viscosity (low shear rate) |
| Shen Yinling 2012 [69] | 33/33 | 19/14 | 18/15 | 65 | 64 | Conventional treatment, Shuxuetong injection 6ml + 0.9% Sodium Chloride Injection 250ml, ivdrip, qd. | Conventional treatment | 15D | Total effective rate TC, TG |
| Zou Yaobing 2012 [70] | 36/36 | 25/11 | 24/12 | 65.1 | 65.1 | Conventional treatment, Shuxuetong injection 6ml + 0.9% Sodium Chloride Injection 250ml, ivdrip, qd. | Conventional treatment | 2W | Plasma viscosity Fibrinogen Whole blood reduced viscosity (high shear rate) Whole blood reduced viscosity (low shear rate) |
| Yang Wenhai 2012 [71] | 50/50 | 29/21 | 27/23 | 40~78 | 39~80 | Conventional treatment, Shenxiongputao injection 100ml, ivdrip, qd. | Conventional treatment | 2W | Total effective rate Plasma viscosity Fibrinogen Whole blood reduced viscosity (high shear rate) Whole blood reduced viscosity (low shear rate) |
| Yang Yi 2017 [72] | 100/100 | 60/40 | 55/45 | 62.9 | 62.3 | Conventional treatment, Shenxiongputao injection 100ml, ivdrip, qd. | Conventional treatment | 7D | Total effective rate Plasma viscosity Fibrinogen Whole blood reduced viscosity (high shear rate) Whole blood reduced viscosity (low shear rate) Incidence of cerebral infarction |
| Geng Wenjing 2013 [73] | 35/35 | 21/14 | 19/16 | 63±4.42 | 62±4.55 | Conventional treatment, Shenxiongputao injection 100ml, ivdrip, qd. | Conventional treatment | 7D | Total effective rate Plasma viscosity Fibrinogen Whole blood reduced viscosity (high shear rate) Whole blood reduced viscosity (low shear rate) Incidence of cerebral infarction |
| Yang Jun 2013 [74] | 58/52 | 36/22 | 31/21 | 60.23 ±9.65 | 58.74 ±10.38 | Conventional treatment, Shenxiongputao injection 100ml, ivdrip, qd. | Conventional treatment | 15D | Total effective rate Plasma viscosity Fibrinogen Whole blood reduced viscosity (high shear rate) Whole blood reduced viscosity (low shear rate) Incidence of cerebral infarction |

(Continued)

**Table 1.** (Continued）

| Included studies | Sample (E/C) | Sex (M/F) E | Sex (M/F) C | Age E | Age C | Intervention E | Intervention C | Disease duration | Outcomes |
|---|---|---|---|---|---|---|---|---|---|
| Li Fengyun 2023 [75] | 46/46 | 24/22 | 23/23 | 64.15 ±3.66 | 63.55 ±3.46 | Conventional treatment, Yinxingyetiquwu injection 6ml + 0.9% Sodium Chloride Injection 250ml, ivdrip, qd. | Conventional treatment | 2W | Plasma viscosity Fibrinogen Adverse events |
| Ma Ling 2018 [76] | 55/55 | 32/23 | 30/25 | 63.17 ±5.79 | 62.38 ±5.74 | Conventional treatment, Yinxingyetiquwu injection 20ml + 0.9% Sodium Chloride Injection 250ml, ivdrip, qd. | Conventional treatment | 2W | Total effective rate Plasma viscosity Fibrinogen Adverse events |
| Li Yuqiong 2018 [77] | 47/47 | 28/19 | 26/21 | 57.43 ±8.54 | 57.36 ±8.41 | Conventional treatment, Yinxingyetiquwu injection 20ml + 0.9% Sodium Chloride Injection 250ml, ivdrip, qd. | Conventional treatment | 2W | Total effective rate. Plasma viscosity Fibrinogen |
| Huang Ping 2017 [78] | 52/52 | 30/22 | 28/24 | 56.8 ±6.3 | 55.2 ±6.1 | Conventional treatment, Yinxingyetiquwu injection 20ml + 0.9% Sodium Chloride Injection 250ml, ivdrip, qd. | Conventional treatment | 2W | Total effective rate. Plasma viscosity Fibrinogen Whole blood reduced viscosity (high shear rate) Whole blood reduced viscosity (low shear rate) Incidence of cerebral infarction |
| Wang Shengping 2016 [79] | 41/41 | 22/19 | 24/17 | 67.2 ±10.1 | 66.8 ±10.3 | Conventional treatment, Yinxingyetiquwu injection 20ml + 0.9% Sodium Chloride Injection 250ml, ivdrip, qd. | Conventional treatment | 2W | Total effective rate. Plasma viscosity Fibrinogen Whole blood reduced viscosity (high shear rate) Whole blood reduced viscosity (low shear rate) |
| Liang Jianwen 2014 [80] | 32/34 | 37/29 | | 43~70 | | Conventional treatment, Yinxingyetiquwu injection 35ml + 0.9% Sodium Chloride Injection 500ml, ivdrip, qd. | Conventional treatment | 10D | Total effective rate. Plasma viscosity Fibrinogen |
| Li Tao 2015 [81] | 34/34 | 41/27 | | 54.3±4.9 | | Conventional treatment, Yinxingyetiquwu injection 70ml + 0.9% Sodium Chloride Injection 250ml, ivdrip, qd. | Conventional treatment | 2W | Total effective rate. Plasma viscosity |

M, male; F, female; E, experimental group; C, control; CT, conventional treament. W, weeks. D, days. Ivdrip, intervenous drop infusion.

-0.27]), Danhong injection + CT (MD = -0.86, 95% CI [-1.33; -0.38]), Shuxuetong injection + CT (MD = -0.89, 95% CI [-1.29; -0.48]), Xuesaitong injection + CT (MD = -0.90, 95% CI [-1.48; -0.33]), and Shuxianin injection + CT (MD = -0.95, 95% CI [-1.42; -0.49]), and there was no significant difference in the comparison between the remaining combination treatments (S5 File).

**3.3.3 Fibrinogen.** Twenty-two studies [34, 38–41, 46, 49, 53, 55, 56, 60, 70–80] (37.93%) and 2314 participants (42.06%) assessed the fibrinogen (Fig 3C). 5 TCM injections combined with CT significantly improved the fibrinogen compared to CT (Fig 4C). Shuxuetong injection (MD = -1.83, 95% CI [-2.77; -0.88]), Dengzhanhuasu injection (MD = -1.31, 95% CI [-2.26; -0.36]), Guhong injection (MD = -1.05, 95% CI [-2.00; -0.10]), Shenxiongputao injection (MD = -0.91, 95% CI [-1.59; -0.22]), and Yinxingyetiquwu injection (MD = -0.71, 95% CI [-1.27; -0.16]) respectively in combination with CT were superior to CT. Ranked by the degree of change in fibrinogen, Shuxuetong injection + CT (P-score = 0.91) was the best and CT (P-

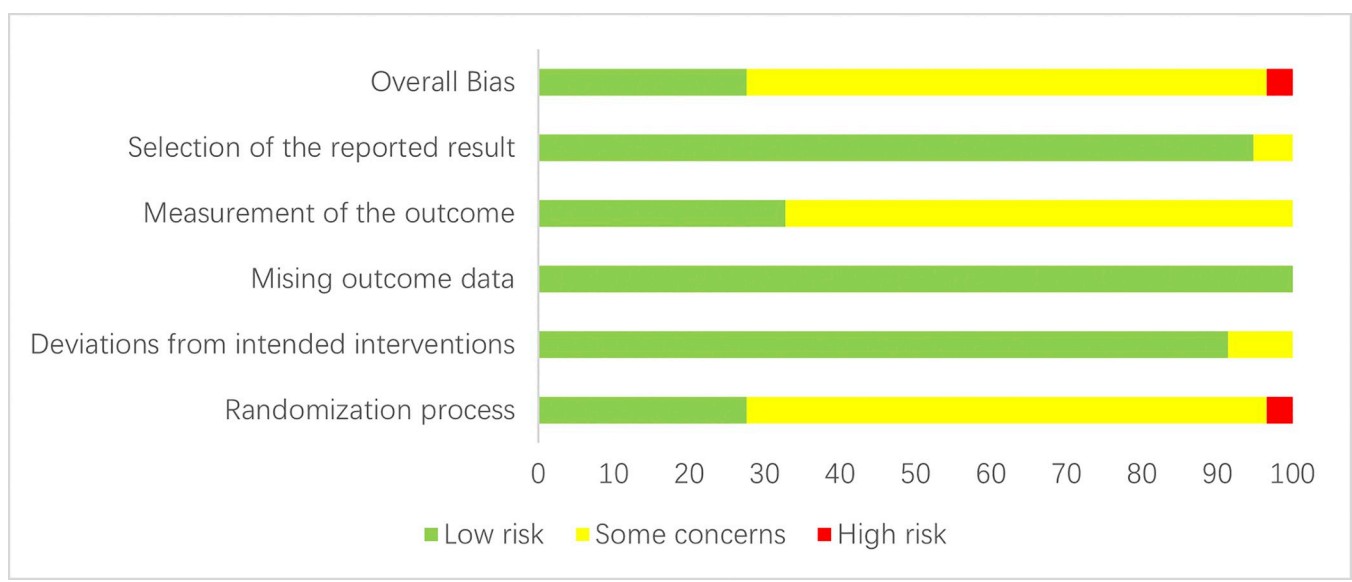

**Fig 2. Results of risk of bias evaluation of included studies.**

score = 0.04) was the worst (Fig 4C). NMA results showed that Shuxuetong injection + CT was superior to Yinxingyetiquwu injection + CT (MD = -1.12, 95% CI [-2.21; -0.02]) and Xue-shuantong injection + CT (MD = -1.34, 95% CI [-2.50; -0.18]), and there was no significant difference between the rest of the combination treatments (S5 File).

**3.3.4 Whole blood reduced viscosity (high shear rate).** Twenty-two studies [29, 30, 34, 46, 49, 53–56, 60, 61, 64, 68, 70–74, 78, 79] (34.48%) and 1898 participants (34.50%) assessed the whole blood reduced viscosity (high shear rate) (Fig 3D). 8 TCM injections combined with CT significantly improved the whole blood reduced viscosity (high shear rate) compared to CT (Fig 4D). Dengzhanhuasu injection (MD = -1.26, 95% CI [-1.67; -0.85]), Yinxingyetiquwu injection (MD = -1.18, 95% CI [-1.58; -0.78]), Xuesaitong injection (MD = -1.03, 95% CI [-1.72; -0.34]), Guhong injection (MD = -0.97, 95% CI [-1.43; -0.52]), Shuxuening injection (MD = -0.90, 95% CI [-1.35; -0.46]), Danhong injection (MD = -0.78, 95% CI [-1.20; -0.37]), Shenxiongputao injection (MD = -0.64, 95% CI [-0.92; -0.36]), and Shuxuetong injection (MD = -0.51, 95% CI [-0.75; -0.26]) respectively in combination with CT were superior to CT. Ranked by the degree of change in whole blood reduced viscosity (high shear rate), Dengzhan-huasu injection + CT (P-score = 0.87) was the best and CT (P-score = 0.07) was the worst (Fig 4D). NMA results showed that Dengzhanhuasu injection + CT was superior to Shuxue-tong injection + CT (MD = -0.75, 95% CI [-1.23; -0.27]) and Shenxiongputao injection + CT (MD = -0.62, 95% CI [-1.12; -0.12]), Yinxingyetiquwu injection + CT was superior to Shuxue-tong injection + CT (MD = -0.67, 95% CI [-1.14; -0.20]) and Shenxiongputao injection + CT (MD = -0.54, 95% CI [-1.03; -0.05]), and there was no significant difference between the rest of the combination treatments (S5 File).

**3.3.5 Whole blood reduced viscosity (low shear rate).** Nineteen studies [29, 30, 34, 46, 49, 53–56, 60, 61, 68, 70–74, 78, 79] (32.76%) and 1828 participants (33.22%) assessed the whole blood reduced viscosity (low shear rate) (Fig 3E). 5 TCM injections combined with CT significantly improved the whole blood reduced viscosity (low shear rate) compared to CT (Fig 4E). Dengzhanhuasu injection (MD = -2.81, 95% CI [-4.58; -1.05]), Shuxuening injection (MD = -2.34, 95% CI [-3.71; -0.96]), Danhong injection (MD = -1.57, 95% CI [-3.00; -0.14]), Shenxiongputao injection (MD = -1.18, 95% CI [-2.09; -0.27]) and Shuxuetong injection (MD

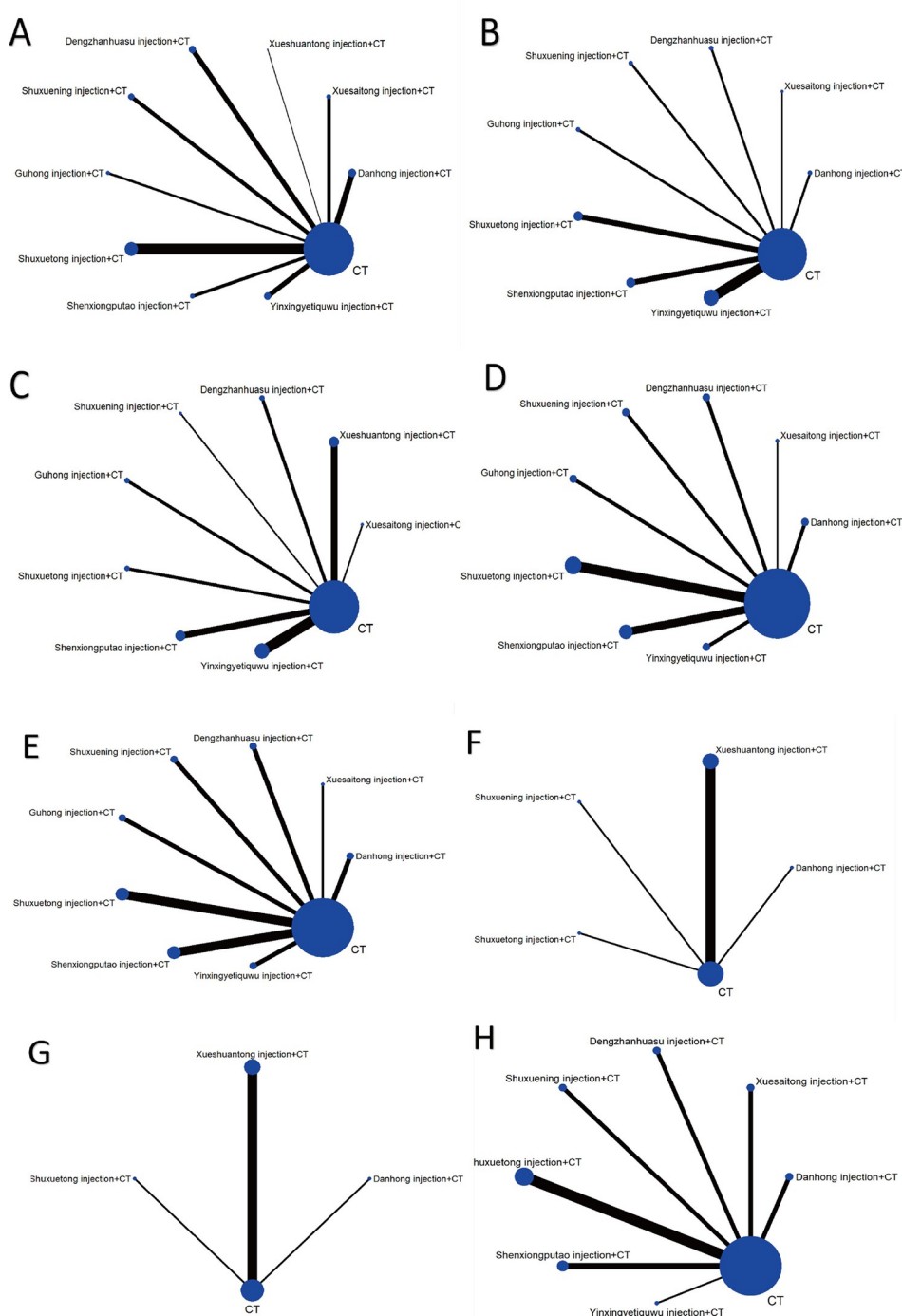

**Fig 3. Network diagrams of comparisons on different outcomes of treatments in different groups of patients with TIA.** (A) total effective rate; (B) Plasma viscosity; (C) Fibrinogen; (D) Whole blood reduced viscosity (high shear rate); (E) Whole blood reduced viscosity (low shear rate); (F) total cholesterol; (G) triglyceride; (H) Incidence of cerebral infarction.

= -1.11, 95% CI [-2.06; -0.16])respectively in combination with CT were superior to CT. Ranked by the degree of change in whole blood reduced viscosity (high shear rate), Dengzhanhuasu injection + CT (P-score = 0.90) was the best and CT (P-score = 0.06) was the worst

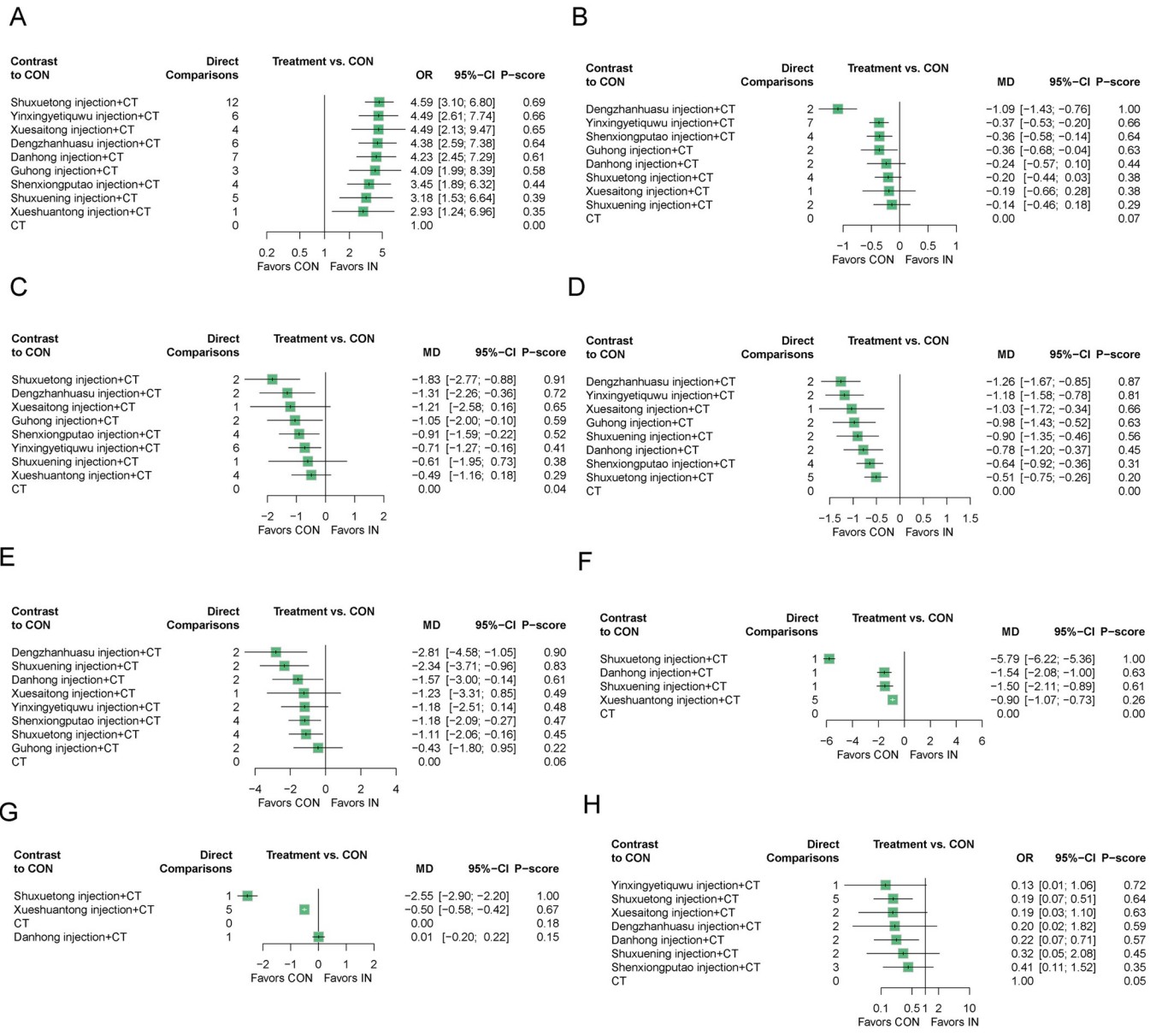

**Fig 4. Forest plots for different outcome indicators.** (A) total effective rate; (B) Plasma viscosity; (C) Fibrinogen; (D) Whole blood reduced viscosity (high shear rate); (E) Whole blood reduced viscosity (low shear rate); (F) total cholesterol; (G) triglyceride; (H) Incidence of cerebral infarction.

(Fig 4E). NMA results showed that Dengzhanhuasu injection + CT was superior to Guhong injection + CT (MD = -2.39, 95% CI [-4.63; -0.15]), and there was no significant difference between the rest of the combination treatments (S5 File).

**3.3.6 Total cholesterol.** Eight studies [30, 38–42, 54, 69] (13.79%) and 808 participants (14.69%) assessed the TC (Fig 3F). 4 TCM injections combined with CT significantly improved the TC compared to CT (Fig 4F). Shuxuetong injection (MD = -5.79, 95% CI [-6.22; -5.36]), Danhong injection (MD = -1.54; 95% CI [-2.08; -1.00]), Shuxuening injection (MD = -1.50, 95% CI [-2.11; -0.89]), and Xueshuantong injection (MD = -0.90, 95% CI [-1.07; -0.73]) respectively in combination with CT were superior to CT. Ranked by the degree of change in whole blood reduced viscosity (high shear rate), Shuxuetong injection + CT (P-score = 1.00)

was the best and CT (P-score = 0) was the worst (Fig 4F). NMA results showed that Shuxue-tong injection was superior to Danhong injection (MD = -4.25, 95% CI [-4.94; -3.56]), Shuxu-ening injection (MD = -4.29, 95% CI [-5.04;-3.54]), and Xueshuantong injection (MD = -4.89, 95% CI [-5.35; -4.43]). Additionally, the Danhong injection + CT was superior to the Xue-shuantong injection + CT (MD = -0.64, 95% CI [-1.20,-0.08]). There was no significant differ-ence between the rest of the combination treatments (S5 File).

**3.3.7 Triglyceride.** Seven studies [30, 38–42, 69] (12.07%) and 746 participants (13.56%) assessed the TG (Fig 3G). 2 TCM injections significantly improved the TG compared to CT (Fig 4G). Shuxuetong injection (MD = -2.55, 95% CI [-2.90; -2.20]) and Xueshuantong injec-tion (MD = -0.50, 95% CI [-0.58; -0.42]) respectively in combination with CT were superior to CT. Ranked by the degree of change in whole blood reduced viscosity (high shear rate), Shux-uetong injection + CT (P-score = 1.00) was the best and Danhong injection (P-score = 0.15) was the worst (Fig 4G). NMA results showed that the Shuxuetong injection + CT was superior to the Xueshuantong injection + CT (MD = -2.05, 95% CI [-2.41; -1.69]). Additionally, the Shuxuetong injection + CT (MD = -2.56, 95% CI [-2.97; -2.15]) and Xueshuantong injection + CT (MD = -0.51, 95% CI [-0.74; -0.29]) were superior to the Danhong injection + CT. There was no significant difference between the rest of the combination treatments (S5 File).

**3.3.8 Incidence of cerebral infarction.** Seventeen studies [26, 28, 35, 36, 46, 48, 53, 54, 61–63, 65, 66, 72–74, 78] (29.31%) and 1702 participants (30.93%) assessed the incidence of cerebral infarction (Fig 3H). 2 TCM injections significantly improved the incidence of cerebral infarction compared to CT (Fig 4H). Shuxuetong injection (OR = 0.19, 95% CI [0.07; 0.51]) and Danhong injection (OR = 0.22, 95% CI [0.07; 0.71]) respectively in combination with CT were superior to CT. Ranked by the degree of change in whole blood reduced viscosity (high shear rate), Yinxingyetiquwu injection + CT (P-score = 0.72) was the best and CT (P-score = 0.05) was the worst (Fig 4H). NMA results showed that there was no significant differ-ence in any of the comparisons between different TCM injections combined with CT (S5 File).

**3.3.9 Cluster analysis.** We performed a cluster analysis of the outcome indicators included in this study to derive the results of two outcome indicators doing well at the same time for different TCM injections. We clustered the total effective rate with plasma viscosity, total effective rate and fibrinogen, total effective rate with whole blood reduced viscosity (high shear rate), total effective rate with whole blood reduced viscosity (low shear rate), total effec-tive rate with incidence of cerebral infarction, and TC with TG, respectively. The clustering of total effective rate and plasma viscosity showed that Dengzhanhuasu injection and Yinxingye-tiquwu injection were located in the upper right corner relatively well (Fig 5A). The clustering of total effective rate and fibrinogen showed that Shuxuetong injection, Dengzhanhuasu injec-tion, and Xuesaitong injection were located in the upper right corner relatively well (Fig 5B). Dengzhanhuasu injection, Yinxingyetiquwu injection, and Xuesaitong injection are relatively better in cluster analyses of total effective rate and whole blood reduced viscosity (high shear rate) (Fig 5C). In the cluster analyses of total efficiency and whole blood reduced viscosity (low shear rate), Dengzhanhuasu injection is relatively better located in the upper right corner (Fig 5D). In the cluster analysis of the total effective rate and incidence of cerebral infarction, Yinxingyetiquwu injection, Shuxuetong injection, Xuesaitong injection, and Dengzhanhuasu were relatively better located in the upper right corner (Fig 5E). In the cluster analysis of lipids, Shuxuetong injection is relatively better located in the upper right corner (Fig 5F).

## 3.4 Adverse reactions

A total of 22 studies [25, 31–33, 37, 40, 51, 54, 59, 63, 64, 67–69, 72–76, 78, 80, 81] reported on the occurrence of adverse reactions involving five types of herbal injections: Danhong

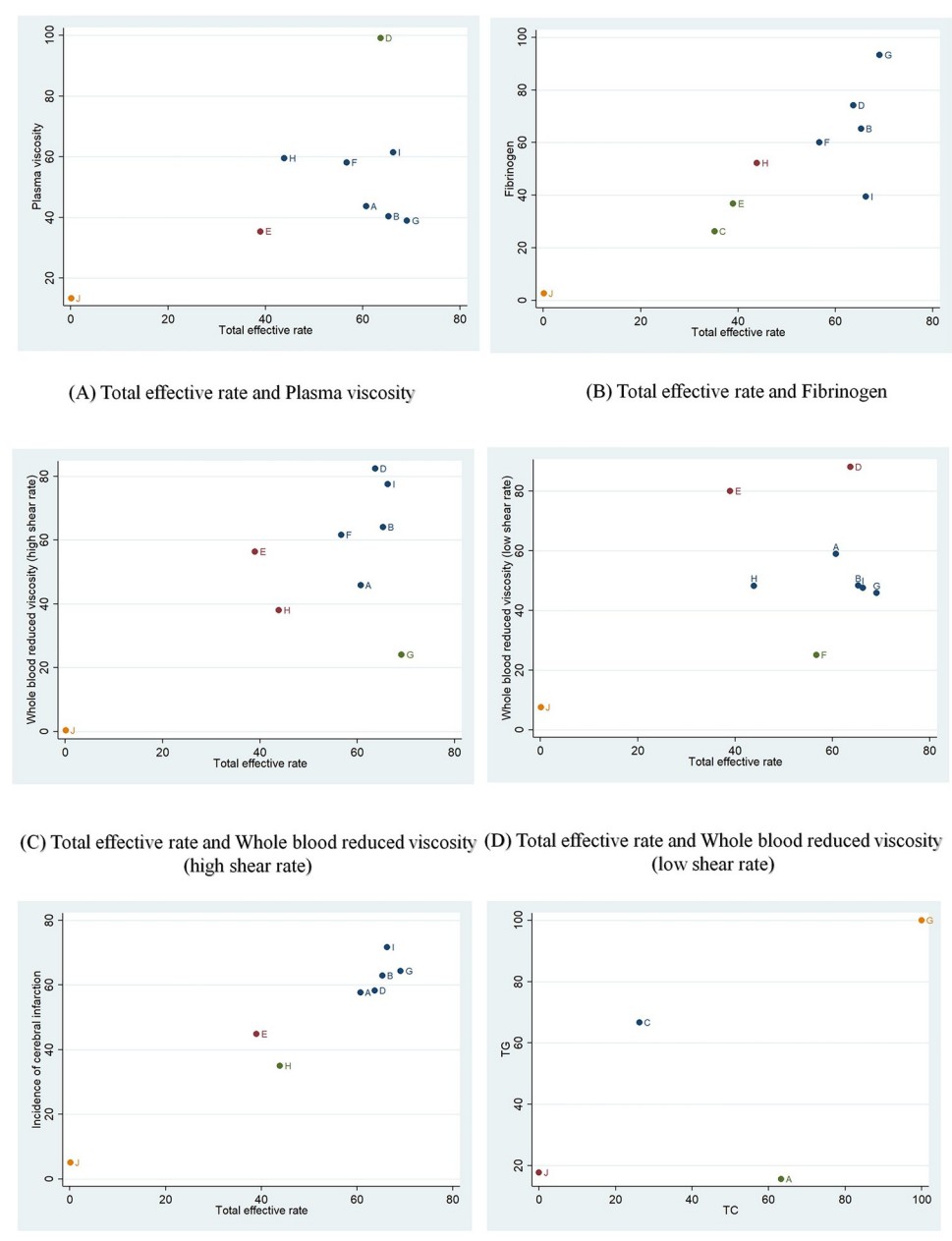

(A) Total effective rate and Plasma viscosity

(B) Total effective rate and Fibrinogen

(C) Total effective rate and Whole blood reduced viscosity (high shear rate)

(D) Total effective rate and Whole blood reduced viscosity (low shear rate)

(E) Total effective rate and Incidence of cerebral infarction

(F) TC and TG

**Fig 5. Cluster analysis of comparisons on different outcomes of treatments in different groups of patients with TIA.** A, Danhong injection+CT; B, Xuesaitong injection+CT; C, Xueshuantong injection+CT; D, Dengzhanhuasu injection+CT; E, Shuxuening injection+CT; F, Guhong injection+CT; G, Shuxuetong injection+CT; H, Shenxiongputao injection+CT; I, Yinxingyetiquwu injection+CT; J, CT.

injection, Xueshuantong injection, Shuxuening injection, Shuxuetong injection, and Yinxingyetiquwu injection. Among these studies, 14 studies [31, 37, 54, 63, 64, 67–69, 72–74, 78, 80, 81] reported no occurrence of adverse reactions, and 8 studies [25, 32, 33, 40, 51, 59, 75, 76] reported the occurrence of adverse reactions. However, only descriptive analyses were performed because the criteria for adverse reactions were not standardized. Specific results are shown in Table 2.

**Table 2. Adverse reactions.**

| Intervention | Number of studies | group | n | Security Information |
|---|---|---|---|---|
| Danhong injection+CT | 2 | E | 9 | abdominal pain 1 case, localised rash 2 cases, loss of appetite 1 case, nausea 2 cases, mild dizziness 1 case, mild liver injury 2 cases |
| | | C | 6 | abdominal pain 1 case, localised rash 1 case, loss of appetite 2 cases, nausea 1 case, mild dizziness 1 case |
| | | E/C | 16 | subcutaneous petechiae 13 cases, gingival bleeding during tooth brushing 3 cases |
| Xueshuantong injection +CT | 1 | E | 3 | dizziness 2 cases, rash 1 case |
| | | C | 2 | dizziness 1 case, rash 1 case |
| Shuxuening injection+CT | 1 | E | 4 | subcutaneous petechiae in 4 cases |
| | | C | 3 | Subcutaneous petechiae in 3 cases |
| Shuxuetong injection+CT | 1 | E | 3 | rash 1 case, nausea 1 case, headache 1 case |
| | | C | 2 | rash 1 case, nausea 1 case |
| Yinxingyetiquwu injection +CT | 2 | E | 5 | loss of appetite 1 case, skin rash 1 case, gastrointestinal discomfort 1 case, hypotension 1 case, headache 1 case |
| | | C | 10 | nausea 2, loss of appetite 3, hypotension 2, headache 2, itchy skin 1 |

E, experimental group; C, control group; CT, conventional treament.

## 3.5 The small sample effect and publication bias

The funnel plot results showed that the total effective rate, plasma viscosity, and fibrinogen funnel plots were basically symmetrical, and the studies were roughly symmetrically distributed on both sides of the midline, suggesting that a small-sample effect is less likely to be present; the funnel plots of the whole blood reduced viscosity (high shear rate), whole blood reduced viscosity (low shear rate), total cholesterol, triglyceride, and incidence of cerebral infarction were less symmetrical, suggesting that there may be a small-sample effect (Fig 6).

## 3.6 Heterogeneity and certainty of evidence

In terms of heterogeneity, heterogeneity was low for total effectiveness rate, TC, TG, and incidence of cerebral infarction, whereas it was high for plasma viscosity, fibrinogen, whole blood reduced viscosity (high shear rate), and whole blood reduced viscosity (low shear rate) (S6 File). The results of the node-splitting method showed that the inconsistency was not significant for all outcomes (S6 File). In addition, for all results, the level of evidence was generally low (S7 File).

## 3.7 Meta-regression and sensitivity analysis

We searched for sources of heterogeneity and tested the stability of the results through meta-regression and sensitivity analyses. The results of the meta-regression showed that for plasma viscosity, an outcome metric, PERIOD was the main reason for its high heterogeneity (Table 3). However, when we reran the analysis with the center values from the meta-regression model, the results were not significantly different from the original results (S8 File). When studies with a "sample size less than 70" were excluded, there was a significant change in the heterogeneity of TG, but this did not have a significant effect on their pooled MD values and rank ordering (S9 File). The sample size may also be a source of heterogeneity in this study. Sensitivity analyses excluding "sample size less than 70" and "less than 10 days and more than 20 days of treatment" did not significantly affect the results, specifically S9 File. Overall, our results were stable.

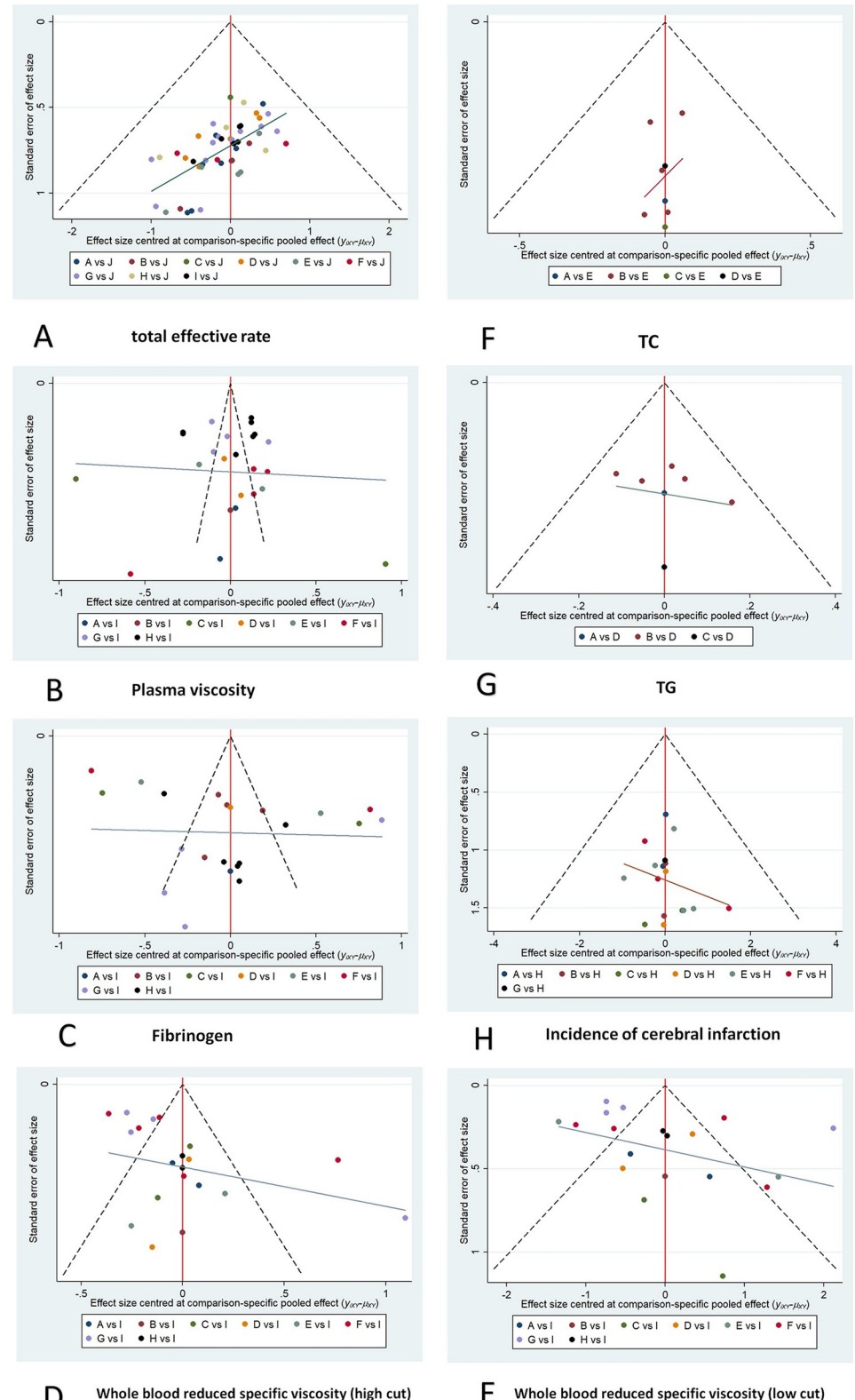

**Fig 6. Comparison-correction funnel chart for ending indicator.** (A) Total effective rate; (B) Plasma viscosity; (C) Fibrinogen; (D) Whole blood reduced viscosity (high shear rate); (E) Whole blood reduced viscosity (low shear rate); (F) Total cholesterol; (G) Triglyceride; (H) Incidence of cerebral infarction.

**Table 3. Network meta-regression.**

| Outcomes | Shared beta (median and 95% CI) | | | | |
|---|---|---|---|---|---|
| | year | sample | male | age | period |
| **Total effectiveness rate** | -0.22 (-0.74; 0.29) | -0.21 (-0.62; 0.21) | -0.20 (-0.60; 0.22) | -0.0004 (-0.46; 0.48) | 0.30 (-0.33; 0.91) |
| **Plasma viscosity** | -0.07 (-0.54; 0.64) | 0.18 (-1.31; 0.56) | 0.15 (-6.04; 0.63) | -0.23 (-0.69; 0.09) | -0.44 (-0.78; -0.09)* |
| **Fibrinogen** | -0.02 (-0.96; 0.80) | 0.02 (-0.63; 0.69) | 0.02 (-0.54; 0.58) | -0.52 (-1.14; 0.20) | -2.57 (-5.65; 0.86) |
| **Whole blood reduced viscosity (high shear rate)** | 0.25 (-0.47; 0.90) | 0.014 (-0.47; 0.52) | 0.002 (-0.50; 0.59) | 0.0007 (-0.47; 0.47) | -0.33 (-0.85; 0.14) |
| **Whole blood reduced viscosity (low shear rate)** | 0.31 (-2.68; 2.16) | 0.18 (-1.43; 2.10) | 0.07 (-1.22; 1.60) | 0.44 (-1.09; 1.89) | -0.98 (-2.27; 0.33) |
| **Total cholesterol** | -0.19 (-2.08; 1.40) | 0.05 (-0.54; 0.56) | 0.04 (-0.79; 0.55) | -0.05 (-0.74; 0.68) | -0.37 (-4.53; 3.59) |
| **Triglyceride** | -0.18 (-1.31; 0.67) | -0.02 (-0.33; 0.40) | -0.02 (-0.44; 0.45) | -0.12 (-0.49; 0.35) | -0.12 (-2.79; 2.20) |
| **Incidence of cerebral infarction** | 0.70 (-0.84; 0.71) | 0.24 (-1.04; 0.24) | 0.18 (-1.20; 1.57) | 0.35 (-1.36; 2.14) | -0.39 (-1.97; 1.15) |

CI: Credible Interval

*: Significant influence factors, 95% CI does not contain zero.

## 4 Discussion

### 4.1 Summary of main findings

Our study encompassed 58 studies related to herbal injections and the findings from 8 studies by using NMA. These findings provided a comprehensive evaluation of the efficacy and safety of different herbal injections when used in conjunction with CT for TIA. Our results revealed the following: (1) Shuxuetong injection exhibited the highest efficacy in improving the total effective rate, TC, TG, and fibrinogen. It ranked second in reducing the incidence of cerebral infarction. (2) Dengzhanhuasu injection emerged as the most effective in reducing plasma viscosity, whole blood reduced viscosity (high shear rate), and whole blood reduced viscosity (low shear rate), and ranked second in reducing fibrinogen, indicating its remarkable efficacy in improving blood hemorheology indicators. (3) Yinxingyetiquwu injection was found to be the most effective in reducing the incidence of cerebral infarction and also demonstrated relative superiority in other metrics. Considering the outcome indicators and cluster analysis results, our study concludes that Shuxuetong injection, Dengzhanhuasu injection, and Yinxingyetiquwu injection all achieved high rankings and hold significant clinical value.

### 4.2 Possible interpretations of the main findings

Chinese medicine has the advantages of multi-components, multi-targets, and multiple pharmacological effects. The combination of Chinese and Western medicine is widely used in the treatment of TIA in China, and the clinical efficacy is remarkable. The nine TCM injections included in the study each demonstrate unique advantages in terms of clinical efficacy. When combined with conventional Western medicine treatments, these injections have different areas of focus for improving TIA indicators. In terms of clinical efficacy, the majority of the included TCM injections possess the ability to promote blood circulation and resolve stasis. Modern pharmacological research has also confirmed their ability to inhibit thrombus formation, exert anticoagulant effects, improve blood circulation, and dilate blood vessels [82]. A table summarizing the mechanisms of TCM injections for TIA is detailed in S10 File.

The primary components of Shuxuetong injection consist of extracts derived from leeches and earthworms, which are used as insect medicines in Chinese traditional medicine due to their potent efficacy in stimulating blood circulation, eliminating blood stasis, and resolving stagnation. They are extensively utilized in the treatment of persistent ailments characterized by the accumulation of "phlegm, blood stasis, and toxicity." Shuxuetong Injection is

considered an optimal solution for reducing blood lipids due to the distinctive effects of leeches and earthworms on lipid absorption and metabolism. Studies have shown that leeches can effectively decrease the synthesis and conversion of fatty acids and cholesterol by inhibiting the expression of critical enzymes, including acyl-coenzyme A-cholesterol acyltransferase, fatty acid synthase, and 3-hydroxy-3-methylglutaryl-coenzyme A reductase (HMGCR) [83]. Extracts derived from earthworms can effectively diminish the absorption of exogenous lipids, inhibit endogenous lipid synthesis, and regulate blood lipid levels [84, 85]. Additionally, clinical reports have suggested the beneficial effects of orally administering leech powder in reducing TC and TG levels among patients with hyperlipidemia [86]. Shuxuetong injection is highly effective in reducing fibrinogen levels, and elevated fibrinogen is closely linked to risk factors for TIA, such as thrombosis and atherosclerosis. According to modern pharmacology, the primary active ingredient found in leeches and earthworms appears to possess potent antithrombotic properties, with a reduced risk of bleeding [87, 88]. In a previous meta-analysis [89], it was determined that the use of leech monotherapy in conjunction with CT significantly enhanced the overall clinical effectiveness and safety in patients with ischemic cerebrovascular disease. However, it should be noted that lipids and fibrinogen were not considered as endpoints in this study. A NMA [90] examining proprietary Chinese medicines derived from leeches for atherosclerosis treatment revealed that these medicines, whether used with or without statins, demonstrated superior efficacy in reducing intima-media thickness and decreasing plaque area and count in comparison to oral statins alone. Of particular significance, both Dilong and leeches exhibit lipid-lowering, antithrombotic, anticoagulant, and anti-inflammatory effects [91–94]. These combined effects could potentially contribute to achieving the highest overall efficacy rate of Shuxuetong injection in TIA treatment while effectively reducing the likelihood of cerebral infarction progression.

The main active ingredient of Dengzhanhuasu injection is derived from the breviscapine extracted from the erigeron breviscapus [95], which contains two main types of Scutellarin and Apigenin. As a kind of Chinese herb, Erigeron breviscapus is widely used in TCM for stroke, paralysis, cold and bruises, etc. It has the effect of activating blood circulation, removing blood stasis and relieving paralysis and pain. In our study, we found that Dengzhanhuasu injection was effective in improving blood rheology indices, which are closely related to vascular traits and blood composition. Previous studies have found that breviscapine can enhance the body's lipid metabolism level by up-regulating the levels of AMP-activated protein kinase (AMPK), p-AMPK, Liver kinase B1 proteins and mRNA, and down-regulating the levels of HMGCR proteins and mRNA, which in turn reduces lipids in the blood [96]. Secondly, it ameliorates atherosclerosis by regulating endothelial cell proliferation and apoptosis through the modulation of Hippo-forkhead box O3A and phosphoinositide 3-kinase/Akt pathway signaling pathways [97]. Breviscapine also inhibits platelet factor 3 and coagulation factor V and significantly increases fibrinolytic activity [98]. More notably, it has been found that breviscapine has neuroprotective effects and that breviscapine slows down brain damage by inhibiting the production of reactive oxygen species, the expression of neuroinflammatory cytokines, and the activation of microglia [99, 100]. These studies have shown that breviscapine improves endothelial cell function and reduces blood lipids, platelets, and other components of the blood, all of which are beneficial in improving blood rheology indices. Meanwhile, its anticoagulant, antiplatelet aggregation, anti-oxidative stress, improvement of atherosclerosis, and protective effects against ischemia/reperfusion [101, 102] have all contributed to the efficacy of Dengzhanhuasu injection in the treatment of TIA.

Ginkgo biloba belongs to Chinese herbal medicine and is widely used to treat cerebrovascular disease, hyperlipidemia, cardiovascular disease, and respiratory disease due to its effectiveness in activating blood circulation, resolving blood stasis, astringing the lungs to calm asthma,

and reducing turbidity and lipid levels. Yinxingyetiquwu injection contains Ginkgo biloba extract as its primary ingredient, which comprises flavonoids, terpenoids, and organic acids [103]. The study found that Yinxingyetiquwu injection, which contains ginkgolides that strongly antagonize platelet-activating factor to prevent thrombus re-formation, effectively reduces the incidence of cerebral infarction [104]. By affecting the metabolism of cyclic AMP, thromboxane A2, and $Ca^{2+}$ in platelets, Ginkgo biloba extract can efficiently inhibit platelet aggregation [105]. Flavonoid components can scavenge superoxide radicals both through endogenous antioxidant systems and directly, contributing to brain protection [106, 107]. Additionally, ginkgolides can activate the Akt/nuclear factor-E2-related factor2 pathway, enhancing cellular antioxidant capacity and attenuating oxidative stress damage [108]. In addition, Ginkgo biloba extract reduces the production of pro-inflammatory cytokines interleukin-1β and tumor necrosis factor-α, increases the levels of anti-inflammatory cytokines interleukin-10 and IL-10R in the brain [109], and reduces cholesterol aggregation in peripheral tissues, thereby providing better protection for vascular endothelial cells [110]. These effects contribute to the improvement of atherosclerosis, subsequently reducing the incidence of TIA or ischemic cerebrovascular disease. Clinical studies have shown that Yinxingyetiquwu injection can reduce the infarct volume caused by stroke, repair neuronal damage [111], and lower blood viscosity [112]. This evidence indicates that Yinxingyetiquwu injection may be effective in improving the incidence of cerebral infarction, total efficiency, and plasma viscosity.

### 4.3 Significance of this study and implications for the future

In the Chinese medical field, TCM injections are becoming a convenient, efficient, and stable medical solution. To the best of our knowledge, this is the first NMA on the efficacy of different TCM injections for the treatment of TIA, which will help clinicians in their decision-making choices. As a carrier of TCM, TCM injections will also contribute to the wide dissemination of TCM in the world. It also provides new treatment ideas for clinicians around the world.

There are still some aspects that need to be noted in future studies. Our study suggests that different herbal injections focus on different therapeutic effects, and further basic experimental studies are needed as to whether this is due to differences in their composition or mechanism. For clinical trials, rational planning of sample size as well as treatment duration is also particularly important. For adverse reactions, studies can be conducted on the composition and dosage of TCM injections. The promotion of TCM injections requires strict production quality, while the quality of both high-quality non-clinical and clinical data will help to make a positive contribution to the development in the international market.

### 5 Limitation

This study has certain limitations, mainly manifested in the following aspects: (1) The majority of the included research literature has a low quality, with 58 studies not clearly specifying the specific randomization methods. (2) The types of TCM injections included in the study were not comprehensive, and all the studies were indirectly compared without direct comparisons. This lack of direct comparison may affect the accuracy and credibility of the results. More head-to-head research is needed to overcome this limitation. (3) There was inconsistency in the treatment period among the included studies, ranging from 1 week to 3 months. Also, meta-regression identified duration of treatment as a source of heterogeneity in this study. (4) The relatively small number of studies Xuesaitong injection (3 cases), Guhong injection (3 cases), and Shenxiongputao injection (4 cases) may have had an impact on the rating of outcome indicators. To overcome this limitation, more high-quality, multi-species TCM

injection-related RCTs need to be included in the analysis to determine their clinical benefits. (5) Most of the literature had limited reporting on adverse reactions, which may affect the assessment of adverse reactions. The reporting of adverse reactions mainly focused on symptom descriptions, while there was limited reporting on safety indicators. For this, studies related to adverse reactions and safety of herbal injections are imminent.

## 6 Conclusion

TCM injections in combination with CT may be a safe and effective intervention for patients with TIA. Current evidence suggests that among the four aspects of improving total effective rate, lowering TC, TG, and fibrinogen, Shuxuetong injection has the best effect. In terms of reducing plasma viscosity, whole blood reduced viscosity (high shear rate), and whole blood reduced viscosity (low shear rate), Dengzhanhuasu injection has the best effect. In terms of reducing the incidence of cerebral infarction, Yinxingyetiquwu injection has the best effect. However, further quantitative analysis is needed to assess the safety aspect. Due to the limitations in the quality and quantity of the included literature, the above conclusions still need to be validated through more high-quality, multicenter, large-sample clinical RCTs.

## Supporting information

**S1 File. PRISMA guideline.**
(DOCX)

**S2 File. Search strategy.**
(DOCX)

**S3 File. Details of the included TCM injections.**
(DOCX)

**S4 File. Risk of bias assessment.**
(DOCX)

**S5 File. Network meta-analysis results.**
(DOCX)

**S6 File. Evaluation of heterogeneity and inconsistency.**
(DOCX)

**S7 File. Grading the evidence for outcomes of the network meta-analysis using CINeMA.**
(DOCX)

**S8 File. Finding sources of heterogeneity using network meta-regression.**
(DOCX)

**S9 File. Sensitivity analyses.**
(DOCX)

**S10 File. Mechanism of action of traditional Chinese medicine injection.**
(DOCX)

## Author Contributions

**Data curation:** Yunhao Yi.

**Formal analysis:** Yunhao Yi.

**Investigation:** Hui Liu.

**Methodology:** Yuanhang Rong.

**Project administration:** Hui Liu.

**Software:** Yunhao Yi, Guangheng Zhang, Yuanhang Rong.

**Supervision:** Shimeng Lv, Yuanhang Rong.

**Writing – original draft:** Yunhao Yi.

**Writing – review & editing:** Ming Li.

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
