## [Decision Letter · Decision Letter 0]

23 May 2024

PONE-D-24-01373Comparative efficacy and safety of traditional Chinese medicine injections in patients with transient ischemic attack: a systematic review and network meta-analysisPLOS ONE

Dear Dr. Li,

Thank you for submitting your manuscript to PLOS ONE. After careful consideration, we feel that it has merit but does not fully meet PLOS ONE’s publication criteria as it currently stands. Therefore, we invite you to submit a revised version of the manuscript that addresses the points raised during the review process.

We look forward to receiving your revised manuscript.

Kind regards,

Elvan Wiyarta, M.D.

Academic Editor

PLOS ONE

Journal Requirements:

This study was supported by a study on the Inheritance Studio of Traditional Chinese Medicine Master Zhang Canjia of State Administration of Traditional Chinese Medicine (no. [2010]59).

Reviewers' comments:

Reviewer's Responses to Questions

**Comments to the Author**

1. Is the manuscript technically sound, and do the data support the conclusions?

Reviewer #1: Yes

Reviewer #2: Partly

Reviewer #3: Yes

2. Has the statistical analysis been performed appropriately and rigorously? 

Reviewer #1: Yes

Reviewer #2: Yes

Reviewer #3: Yes

3. Have the authors made all data underlying the findings in their manuscript fully available?

Reviewer #1: Yes

Reviewer #2: Yes

Reviewer #3: Yes

4. Is the manuscript presented in an intelligible fashion and written in standard English?

Reviewer #1: Yes

Reviewer #2: Yes

Reviewer #3: Yes

5. Review Comments to the Author

**Reviewer #1:** Upon reviewing the manuscript titled " Comparative efficacy and safety of traditional Chinese medicine injections in patients with transient ischemic attack: a systematic review and network meta-analysis," I offer the following comments and suggestions:

1. Regarding the Introduction Section and Expression：

The author’s eloquent language and excellent command of English are praiseworthy. The author provided a clear explanation of the definition, epidemiology, and shortcomings of conventional Western medical treatment for transient ischemic attacks. Subsequently, the author emphasized the advantages of traditional Chinese medicine injections and highlighted the current research limitations in this area. In the final paragraph, the author discussed how the integration of traditional Chinese medicine with Western medicine can improve cure rates and reduce the recurrence or occurrence of strokes. However, the author does not specify whether results were obtained in animal experiments or if various traditional meta-analyses have validated this conclusion. Therefore, relying solely on the results of one meta-analysis or directly describing its clinical effects may lack persuasive strength. It is recommended to initially outline the advantages of traditional Chinese medicine injections, followed by a discussion on the results and shortcomings of prior meta-analyses, leading to the introduction of the study’s objectives. Including an abbreviation table in the supplementary material or at the end of the article is advisable.

2. Clarity of Research Design and Methodology:

This study systematically reviewed the effects of various traditional Chinese medicine injections on transient ischemic attack through a network meta-analysis approach. Particularly noteworthy is the incorporation of literature from various databases, including four Chinese databases, thereby enhancing the study’s scope and diversity. The authors are recommended to detail specific search strategies in the methodology section, such as the PubMed search strategy. Additionally, the statement “The results of these analyses are shown in the table below” in Section 2.7 Statistical Investigation may not be contextually relevant or may be inaccurately formulated. It was noted that the authors employed cluster analysis, and it is advisable to explain this in Section 2.7 Statistical Investigation.

3. Interpretation and Application of Results:

The use of Stata software and a random-effects model in the analysis is commendable for its rigor. The research results indicate that the combination of traditional Chinese medicine injections with conventional treatment is an effective and safe method for treating transient ischemic attacks. The discussion section is logically structured and the results are convincing. In conclusion, it would be beneficial to explore how these findings may impact future research.

In the discussion section, the author’s description of the mechanisms is comprehensive, linking the research results with the mechanisms. However, I suggest organizing the discussion of the mechanisms in the order of research conclusions (1), (2), (3), (4). Additionally, it would be advisable to discuss the positive outcome indicators in separate paragraphs while maintaining logical coherence within the context.

4. Discussion on Limitations and Future Research Directions:

While the article has already mentioned some research limitations, such as variations in treatment duration and a limited number of studies, it is recommended to further investigate strategies for addressing these limitations and to explore novel methodologies or technologies that could be utilized in future research.

Overall, this manuscript offers valuable insights into the efficacy and safety of traditional Chinese medicine injections for treating transient ischemic attacks. Its strength lies in a comprehensive approach and rigorous analysis, making a significant contribution to the field. Implementing the above suggestions can further enhance the clarity and impact of the manuscript.

**Reviewer #2:** Please follow the PRISMA 2020 statement.

Please conduct adequate subgroup and sensitivity analysis to test the robustness of the results.

There is no application of the GRADE approach.

The sources of heterogeneity or factors influencing the meta-analysis results are not properly explored or explained. Please provide a detailed and thorough discussion on the potential sources of heterogeneity or factors affecting the meta-analysis results.

An appealing review should include a critical assessment of the relevant literature published.

Can you elaborate on the importance and real-world implications of your research findings? Additionally, can you suggest any future research directions to explore?

Please improve the English writing throughout the manuscript.

**Reviewer #3: **This manuscript is interesting and contributes to the field, however, to increase readability and facilitate reader understanding, graphical figures are needed on the mechanistics of traditional Chinese medicine injections in the improvement of patients with transient ischemic attacks.

6. PLOS authors have the option to publish the peer review history of their article (what does this mean?). If published, this will include your full peer review and any attached files.

Reviewer #1: No

Reviewer #2: No

Reviewer #3: No

---

## [Author Response · Author response to Decision Letter 0]

17 Jun 2024

Dear Editor and Reviewers,

Thank you very much for giving us opportunities to revise our manuscript, and we appreciate the reviewer a lot for his positive and constructive comments and suggestions. We have studied reviewer’s comments carefully and have made revisions, which are highlighted in the paper. We hope the corrections will meet with your approval.

We corrected the formatting and wording of the article according to the journal's requirements. In addition, we had to use R software to respond to some of the reviewers' comments, which added a lot of figures and tables as well as textual descriptions, but this was very beneficial to the quality of our article. We changed our funding sources in the cover letter. We added the doi of the citation, and we changed the file name according to the journal's requirements. In addition, we modified some of the graphs and changed the graphs to TIFF format with higher quality images.

Reviewer #1: Upon reviewing the manuscript titled " Comparative efficacy and safety of traditional Chinese medicine injections in patients with transient ischemic attack: a systematic review and network meta-analysis," I offer the following comments and suggestions:

1. Regarding the Introduction Section and Expression：

The author’s eloquent language and excellent command of English are praiseworthy. The author provided a clear explanation of the definition, epidemiology, and shortcomings of conventional Western medical treatment for transient ischemic attacks. Subsequently, the author emphasized the advantages of traditional Chinese medicine injections and highlighted the current research limitations in this area. In the final paragraph, the author discussed how the integration of traditional Chinese medicine with Western medicine can improve cure rates and reduce the recurrence or occurrence of strokes. However, the author does not specify whether results were obtained in animal experiments or if various traditional meta-analyses have validated this conclusion. Therefore, relying solely on the results of one meta-analysis or directly describing its clinical effects may lack persuasive strength. It is recommended to initially outline the advantages of traditional Chinese medicine injections, followed by a discussion on the results and shortcomings of prior meta-analyses, leading to the introduction of the study’s objectives. Including an abbreviation table in the supplementary material or at the end of the article is advisable.

Response: Many thanks to the reviewer for their recognition and comments, which will make our exposition clearer and more organized. We outlined the advantages of herbal injections, giving examples of experimentally relevant findings of one type of herbal injection. Since we did not find traditional meta and reticulated meta for the treatment of transient ischemic attack with herbal injections during the search, we did not discuss the results and shortcomings of the previous meta-analysis. We therefore added the results of this study in S5 file.The results of the network meta-analysis are shown in the lower left part, and results from pairwise comparisons in the upper right half (if available). We rechecked the full text for acronyms and added acronyms in the S11 file. Thanks again for your comments.

2. Clarity of Research Design and Methodology:

This study systematically reviewed the effects of various traditional Chinese medicine injections on transient ischemic attack through a network meta-analysis approach. Particularly noteworthy is the incorporation of literature from various databases, including four Chinese databases, thereby enhancing the study’s scope and diversity. The authors are recommended to detail specific search strategies in the methodology section, such as the PubMed search strategy. Additionally, the statement “The results of these analyses are shown in the table below” in Section 2.7 Statistical Investigation may not be contextually relevant or may be inaccurately formulated. It was noted that the authors employed cluster analysis, and it is advisable to explain this in Section 2.7 Statistical Investigation.

Response: Many thanks to the reviewers for their careful suggestions. Since the search terms are too cumbersome and may take up a lot of space in the main text, we have added some of the main search terms in the main text. And we put the main English search terms in the appendix. We deleted irrelevant statements. In addition, because of the inclusion of heterogeneity, meta-regression, sensitivity analysis, etc., we made some additions and corrections to section 2.7. Importantly, we also added a description of cluster analysis. Thanks again for your suggestions.

3. Interpretation and Application of Results:

The use of Stata software and a random-effects model in the analysis is commendable for its rigor. The research results indicate that the combination of traditional Chinese medicine injections with conventional treatment is an effective and safe method for treating transient ischemic attacks. The discussion section is logically structured and the results are convincing. In conclusion, it would be beneficial to explore how these findings may impact future research.

Response: Thank you for your suggestions for us, which will lead to a more organized discourse. We have streamlined our findings and replaced the order of the exposition with a more organized discussion of positive outcome indicators in segments.

Reviewer #2:

Please follow the PRISMA 2020 statement.

Please conduct adequate subgroup and sensitivity analysis to test the robustness of the results.

There is no application of the GRADE approach.

The sources of heterogeneity or factors influencing the meta-analysis results are not properly explored or explained. Please provide a detailed and thorough discussion on the potential sources of heterogeneity or factors affecting the meta-analysis results.

An appealing review should include a critical assessment of the relevant literature published.

Can you elaborate on the importance and real-world implications of your research findings? Additionally, can you suggest any future research directions to explore?

Please improve the English writing throughout the manuscript.

Response: 

We thank the reviewer for comments, which were very helpful in improving the overall quality of our study.

First we complied with the PRISMA 2020 statement by adding an exploration of sensitivity analysis and heterogeneity correlation. At the same time, we re-supplemented the PRISMA 2020 statement in the Appendix. We performed meta regression to find the source of heterogeneity and reran the NETWORK meta analysis to test the stability of the results based on the center value of the model. We reran the sensitivity analysis to compare with the original results, in addition we observed the change of heterogeneity.

Notably, we used R software to explore heterogeneity, perform meta-regressions and test the stability of the results using their center values. This made it necessary for us to change the software. But our rank ordering remained the same. There were only minor changes in OR or MD values and their 95% CIs, which were unavoidable with each reanalysis.

We utilized CINeMA for level of evidence evaluation. CINeMA (Confidence In Network Meta-Analysis: http://cinema.ispm.ch/) is a web application that builds upon GRADE (Grading of Recommendations Assessment, Development and Evaluation). It assesses the certainty of evidence across six domains: within-study bias, reporting bias, indirectness, imprecision, heterogeneity, and incoherence. By using these criteria to rate the certainty of evidence, CINeMA improves the transparency, reproducibility, and credibility of research results.

We strongly agree that an attractive review should include a critical assessment of the published relevant literature. Unfortunately, our search process did not reveal traditional meta-analyses and NMAs of herbal injections for the treatment of TIA, and therefore we were unable to compare them with published studies . However, we interspersed the narrative in 4.2 with explorations of previous studies. Not only that, we added two kinds of results in S5 file.The results of the network meta-analysis are shown in the lower left part, and results from pairwise comparisons in the upper right half (if available). This is a side note to the importance of our study.

We have added the importance and real-world impact of the findings in section 4.3 and made some observations on the future research direction of Chinese medicine injection. In addition, in the process of overall supplementation and revision, we also focused on the English writing of the whole manuscript.

Once again, we would like to thank the reviewer for painstaking suggestions to improve the quality of our study.

Reviewer #3: This manuscript is interesting and contributes to the field, however, to increase readability and facilitate reader understanding, graphical figures are needed on the mechanistics of traditional Chinese medicine injections in the improvement of patients with transient ischemic attacks.

Response: 

We thank the reviewer for valuable suggestions to improve the readability of our study. We have shown the mechanism of action of different injections in the form of graphs and tables in the S10 File. Once again, we are grateful for your valuable suggestions.

---

## [Decision Letter · Decision Letter 1]

10 Jul 2024

Comparative efficacy and safety of traditional Chinese medicine injections in patients with transient ischemic attack: a systematic review and network meta-analysis

PONE-D-24-01373R1

Dear Dr. Li,

We’re pleased to inform you that your manuscript has been judged scientifically suitable for publication and will be formally accepted for publication once it meets all outstanding technical requirements.

Kind regards,

Elvan Wiyarta, M.D.

Academic Editor

PLOS ONE

Additional Editor Comments (optional):

Reviewers' comments:

Reviewer's Responses to Questions

**Comments to the Author**

1. If the authors have adequately addressed your comments raised in a previous round of review and you feel that this manuscript is now acceptable for publication, you may indicate that here to bypass the “Comments to the Author” section, enter your conflict of interest statement in the “Confidential to Editor” section, and submit your "Accept" recommendation.

Reviewer #2: All comments have been addressed

Reviewer #3: All comments have been addressed

2. Is the manuscript technically sound, and do the data support the conclusions?

Reviewer #2: Yes

Reviewer #3: Yes

3. Has the statistical analysis been performed appropriately and rigorously? 

Reviewer #2: Yes

Reviewer #3: Yes

4. Have the authors made all data underlying the findings in their manuscript fully available?

Reviewer #2: Yes

Reviewer #3: Yes

5. Is the manuscript presented in an intelligible fashion and written in standard English?

Reviewer #2: Yes

Reviewer #3: Yes

6. Review Comments to the Author

Reviewer #2: (No Response)

Reviewer #3: (No Response)

7. PLOS authors have the option to publish the peer review history of their article (what does this mean?). If published, this will include your full peer review and any attached files.

Reviewer #2: **Yes: **Yongliang Jia

Reviewer #3: No

---

## [Editor Report · Acceptance letter]

15 Jul 2024

PONE-D-24-01373R1 

PLOS ONE

Dear Dr. Li, 

I'm pleased to inform you that your manuscript has been deemed suitable for publication in PLOS ONE. Congratulations! Your manuscript is now being handed over to our production team.

Kind regards, 

on behalf of

Mr. Elvan Wiyarta 

Academic Editor

PLOS ONE